# Molecular tension indicators reveal unexpectedly complex regulation of tension in live mouse organs

Keita Fujiwara[1,2], Katsunori Fujiki[3], Tomoya O. Akama[1], Katsuhiko Shirahige[3,4], Ichiro Shiojima[2], Tomoyuki Nakamura[1] & Maretoshi Hirai [1] ✉

Since the emergence of molecular tension sensors, the understanding of mechanical forces has advanced substantially. However, visualizing molecular tension in mammalian tissues has remained challenging owing to the technical constraints of the Förster resonance energy transfer (FRET)-based molecular tension sensors. Here, we develop a molecular tension sensor based on circularly permuted EGFP with an elastic linker, which is inserted into either αActinin or αCatenin and then fused with mCherry at the C-terminus to simultaneously visualize tension and the amount of sensor protein. This single-fluorophore tension indicator enables subtle tension changes to be visualized simply from color tone under superresolution microscopy. We further generate H11 knock-in mice expressing these indicators, revealing a molecular-specific regulation of mechanical load within tissues. Thus, our molecular tension indicators provide a powerful approach for probing the complex and heterogeneous regulation of mechanical forces in vivo mammalian systems.

Mechanical forces play critical roles in embryogenesis, maintaining homeostasis, and pathological processes. Understanding the mechanical aspects of biological processes is essential, and the field of mechanobiology has been expanding. In particular, since molecular tension sensors (TSs) were developed and enabled tension visualization in living cells[1], the understanding of intracellular tension has accelerated. These TSs are based on Förester resonance energy transfer (FRET), in which two fluorophores flank an extensible linker inserted into a deformable region of tension-sensing molecule. The original vinculin TS used a spider-silk-derived $(GPGGA)_n$ linker[1], and subsequent work has introduced additional extensible linkers with distinct mechanical properties and expanded dynamic ranges[2]. In combination with a variety of cytoskeletal or junctional proteins, numerous molecular TSs have been developed thus far[3–8], and have provided important insights into mechanotransduction.

FRET-based TSs have also been successfully applied in vivo[9–11]. Although these studies highlight the power of FRET-based approaches, their performance can vary across systems; for example, the tension detected by TSs was reported not to reflect the expected tension pattern or to respond to mechanical perturbations in certain tissues[12]. Several considerations need to be accounted for when utilizing FRET-based TSs[13]: (1) FRET efficiency depends on both donor–acceptor distance and the relative orientation of their transition dipoles; (2) quantitative FRET analysis requires optical

corrections such as bleed-through compensation; and (3) although acceptor excitation can quantify TS abundance, comparing TS levels across heterogeneous in vivo tissues is challenging due to differences in expression levels and optical environments among cell types. These issues prompted us to develop a tension indicator to explore the physical aspects of complex in vivo situations.

For this purpose, we adopted an approach with circular permutation of green fluorescence protein (GFP)[14] to insert an elastic linker. This approach has been successfully applied to G-CaMP, which is a genetically encoded calcium indicator and has been refined over time by incorporating mutations into circularly permuted EGFP (cpEGFP)[15,16]. In this study, we utilized cpEGFP with mutations applied to G-CaMP8[16] and the linker of spider silk $(GPGGA)_n$ to develop a new TS, enabling visualization of tension with a single fluorophore. This TS was inserted into the deformable region of αActinin and αCatenin, which have been well characterized as tension transducers and applied to conventional TSs[4,5,8,17]. Each hybrid protein was subsequently fused with mCherry at the carboxy-terminal (C-terminal) end to monitor the amount of our TS proteins simultaneously with tension force. These tension indicators enabled us to visualize the dynamics of tension with unprecedentedly high resolution and high sensitivity. Furthermore, we generated the gene-targeting mice that express either the αActinin or αCatenin tension indicator. These analyses revealed that tension

[1]Department of Pharmacology, Kansai Medical University, Hirakata, Osaka, Japan. [2]Department of Medicine II, Kansai Medical University, Hirakata, Osaka, Japan. [3]Laboratory of Genome Structure and Function, Institute for Quantitative Biosciences, The University of Tokyo, Tokyo, Japan. [4]Laboratory of Chromosome dynamics and genome stability, Department of Cell and Molecular Biology, Karolinska Institutet, Stockholm, Sweden. ✉e-mail: hirai.mar@kmu.ac.jp

is substantially different depending on which molecule is used to detect even at the same locus, suggesting that tension relies not on a single molecule but on multiple molecules in a distinctive way through interactions with several other molecules. These results indicate that the regulation of tension force in vivo is far more complex than previously considered. Thus, our tension indicators provide a path for a deeper understanding of mechanical phenomena in biological processes.

## Results

### Development of the αActinin and αCatenin TS indicators with cpEGFP

Most previously reported TSs have relied on FRET systems[3–8]. Although powerful, FRET-based sensors can be difficult to interpret in complex tissue environments because of their sensitivity to optical and biochemical conditions. To address these limitations, we developed a single-fluorophore cpEGFP-based tension indicator that reports force through deformation of the β-barrel structure rather than changes in donor–acceptor energy transfer. Thus, we started with the modification of a single fluorophore, EGFP, to detect tension without FRET. Two types of TSs were generated: (i) TS-1, a simple insertion of an elastic linker $(GPGGA)_8$ at Tyr145 of EGFP, where GFP has been shown to tolerate the insertion of the full protein[1,14], and (ii) TS-2, cpEGFP with the same elastic linker[14] (Fig. 1A). Here, we utilized an optimized cpEGFP of G-CaMP8[16]. For analysis of the fluorescence intensity of these TSs, either TS-1 or TS-2 was inserted between the SR1 and SR2 domains of αActinin and then fused with mCherry at the C-terminal end. Unexpectedly, only MDCKII cells expressing ActTS-2 emitted green fluorescence, while those expressing ActTS-1 did not (Fig. 1B). Therefore, TS-2 was henceforth employed as a TS, and ActTS-2 fused with mCherry at the C-terminus was designated as an αActinin tension indicator.

Among the molecules that have been applied to TSs[3–8], two intracellular force-bearing proteins, αActinin and αCatenin have been most extensively characterized[5,8,17–19]. These molecules localize to distinct subcellular regions, such as actin filaments and adherens junctions (Supplementary Fig. 1A), and therefore are expected to report different aspects of intracellular tension. To visualize tension acting on αCatenin, we inserted the TS in place of the M region and fused with mCherry to the C-terminus, thereby generating αCatenin tension indicator. As expected, the fluorescence of the αCatenin tension indicator was localized predominantly at cell margins (Supplementary Fig. 1B).

In these tension indicators, the green fluorescence intensity of the TS diminishes when tension increases, whereas the red fluorescence intensity remains constant; thus, the tension indicator presumably shifts from yellow-green to orange-red (Fig. 1C). To determine whether our tension indicator works as expected, we used the myosin II inhibitor, blebbistatin. As shown in Fig. 1D, blebbistatin decreases tension, which should increase the green fluorescence of TS. As expected, the fluorescence of the αActinin tension indicator changed from orange to yellow-green in response to blebbistatin (Fig. 1E). Furthermore, the green fluorescence from αActinin TS increased significantly in a dose-dependent manner in response to blebbistatin in MDCKII cells, as did the relaxation ratio (green/red fluorescence ratio) (Fig. 1F and Supplementary Fig. 1C), without detectable bleaching effects (Supplementary Fig. 1D).

For determination of the range of force that the TS could detect, the TS protein was purified and conjugated to oligo-arms (Fig. 1G, H, and Supplementary Fig. 1E). After biotin-conjugated 5.0 kb handle DNAs were annealed to the TS with oligo-arms, the force, distance, and fluorescence intensity of single TS molecules were measured using optical tweezers and fluorescence microscopy (Fig. 1I–K). As shown in Fig. 1J, K, the fluorescence intensity of the TS changes with force between 0 and 6 pN. Importantly, the TS exhibited highly reversible mechanical behavior: the same molecule produced nearly identical force–distance curves during three consecutive loading–unloading cycles between 0 and 10 pN (Supplementary Fig. 1F). Overall, these results indicate that our αActinin tension indicator detects changes in intracellular tension.

### Detection of intracellular tension on αActinin

To verify how effectively our TS detects intracellular tension, we examined several control mutants in MDCKII cells (Fig. 2A and Supplementary Fig. 2A). As shown, the original αActinin tension indicator significantly increased the relaxation ratio in response to blebbistatin (Figs. 1F and 2A). In contrast, Control-1 (αAct TS ind with flexible linker), in which the elastic linker was replaced with the GGTGGS linker, showed a smaller and slower increase. Control-2 (αAct EGFP), in which the TS module was replaced with wild-type EGFP, showed no change. Control-3 (ΔABD αAct TS ind), lacking the actin binding domain, exhibited only a minimal increase. For reference, the relative brightness of αAct-TS and αAct-EGFP, as well as the normalization procedure, are shown in Supplementary Fig. 2B. Consistently, in response to para-amino-blebbistatin (p-AmBleb), a myosin inhibitor that is not light sensitive[20], the αActinin tension indicator again showed a significant increase in relaxation ratio (Supplementary Fig. 2C, D) without detectable photobleaching (Supplementary Fig. 2E), whereas αAct EGFP showed little or no change (Supplementary Fig. 2C, D). Together, these results demonstrate that our TS module detects intracellular tension with the highest sensitivity among the constructs tested.

Next, we examined whether our indicator is sensitive enough to detect the intracellular difference in tension. For this purpose, we first examined the distribution of actin filaments in NIH3T3 cells by phalloidin staining, which revealed that their distribution varies with height (Fig. 2B). The z-stack images were divided into four substacks by height, color-coded, and reconstructed as a 3D image (Fig. 2C). It clearly delineated that the distribution of actin filaments is considerably different along with the contour of the cell (Fig. 2C, bottom), indicating that tension on αActinin could vary with height as well.

As expected, the merged images of αActinin-TS and αActinin-mCherry showed distinct color tones depending on height (Fig. 2D, left, Pre). The merged images appear orange-red at the bottom and yellow-green at the top, indicating that αActinin at the bottom was under greater tension, whereas αActinin at the top was more relaxed. The relaxation-ratio images (green/red ratio) visualized with a cool-warm colormap provide a quantitative view of this difference, making the height-dependent variation in tension more apparent than the merged images (Fig. 2D, right). After the addition of blebbistatin, tension on αActinin was strongly relaxed in the middle of the cells, whereas it was not considerably changed at the bottom. The relaxation-ratio images also clearly highlight these changes, further demonstrating that our αActinin tension indicator is sensitive enough to detect the intracellular differences in tension.

### Detection of tension dynamics with the αActinin and αCatenin tension indicators

Having established the performance of the αActinin tension indicator, we next evaluated whether our αCatenin TS indicator similarly detects intracellular tension, because αCatenin experiences actin-transmitted forces. To assess how the αCatenin tension indicator responds to intracellular tension, we examined several control mutants in NIH3T3 cells (Fig. 3A and Supplementary Fig. 3A). As shown, the original αCatenin tension indicator exhibited a significant increase in the relaxation ratio upon blebbistatin treatment, whereas αCat EGFP lacking the TS linker showed no detectable change. A second control (Control-2; ΔCtnnb BD αCat TS ind), which lacks the β-catenin binding domain, exhibited markedly diminished responses. Because α-catenin predominantly functions as a 1:1 heterodimer with β-catenin[21,22], this reduced response indicates that the αCatenin TS indicator reports tension specifically in the β-catenin engaged, force-transmitting state. Consistent with these findings, the αCatenin tension indicator also showed a significant increase in relaxation ratio following para-amino-blebbistatin (p-AmBleb) treatment, whereas αCat EGFP showed little or no change (Supplementary Fig. 3B, C).

Having validated both tension indicators, we next investigated how they report tension dynamics in living cells. We used the αActinin tension indicator to visualize forces in lamellipodia, which drive cell movement, and

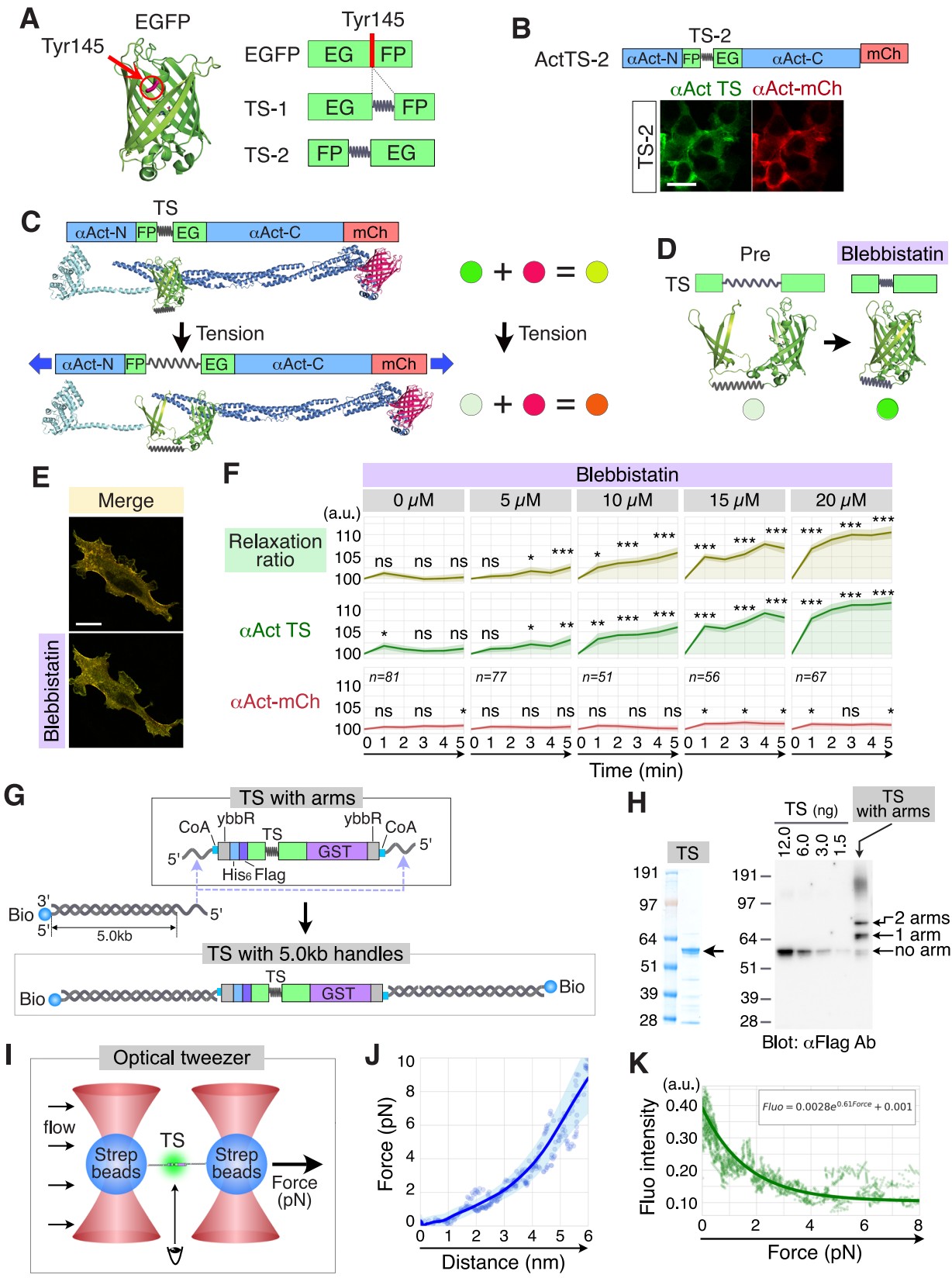

in filopodia, which guide direction. Although both structures rely on actin filaments, tension acting on αActinin in these structures has not been visualized previously. We therefore performed time-lapse imaging of NIH3T3 cells expressing the αActinin tension indicator. As shown in Fig. 3B (solid rectangle) and 3C, lamellipodia were yellow-green in the merged

image and warm color in terms of the relaxation ratio, indicating that tension was relaxed in lamellipodia (Supplementary Video 1). We quantitated the fluorescence intensity along the leading edge (Fig. 3C; dotted line) and calculated the relaxation ratio. Tension dynamics were visualized as a heatmap by expanding the relaxation ratio on the time axis and the position

**Fig. 1 | Development of the αActinin tension indicators with cpEGFP. A** The position of Tyr145 in the crystal structure of EGFP. Tension sensor-1 (TS-1): an elastic linker (GPGGA)$_8$ was inserted in-frame at Tyr145 of EGFP. Tension sensor-2 (TS-2): an optimized cpEGFP of GCaMP8 was utilized. A flexible linker was replaced with an elastic linker (GPGGA)$_8$. **B** Schematic illustration of αActinin tension indicator, ActTS-2. Either TS (TS-1 or TS-2) was inserted between the SR1 and SR2 domains of αActinin in ActTS-1 and ActTS-2, respectively. SR: spectrin repeat domain. To monitor the amount of tension indicator, each αActinin fused with TS was fused to mCherry at the C-terminus. Representative fluorescence microscopy images of NIH3T3 cells show that while ActTS-1 does not show any green fluorescence, ActTS-2 shows green fluorescence. Scale bar: 20 μm. **C** Schematic illustration of the αActinin tension indicator with or without tension, and the predicted fluorescence change with tension. TS: tension sensor. **D** Schematic illustration of the predicted change in the fluorescence of TS resulting from the addition of the myosin II inhibitor blebbistatin. **E** Fluorescence microscopy of NIH3T3 cells expressing the αActinin tension indicator before and after the addition of blebbistatin. Scale bar: 20 μm. **F** Line plots of time-lapse fluorescence microscopy images of MDCKII cells expressing the αActinin tension indicator before and after the addition of different concentrations of blebbistatin. Shaded areas indicate 95% confidence intervals (CIs). The exact sample size for each condition is shown directly on the plots ($n$ = number of cells). Statistical analysis was performed using a one-way ANOVA, followed by the Scheffé post hoc test; ns: $p > 0.05$, *: $p < 0.05$, **: $p < 0.01$, ***: $p < 0.001$. Asterisks indicate statistical significance at 1, 3, and 5 min compared to 0 min. **G** Schematic illustration of the assembly strategy for conjugating the TS with 5-kb DNA handles for single-molecule experiments. **H** Coomassie blue staining and Western blotting of purified TS with or without oligo-arms. **I** Schematic illustration of the optical-tweezer setup used to apply force to a single TS molecule. **J** Force–extension curves obtained from single TS molecules. **K** Relationship between applied force and TS fluorescence intensity, fitted with the equation $Fluo = 0.0028 \times e^{0.61 \times Force} + 0.001$.

axis (Fig. 3D). The heatmap revealed that αActinin tension in lamellipodia is constantly changing without any regularity. In contrast, the filopodia were orange-red in the merged image (Fig. 3B, dotted rectangle) The cool blue areas of the tension ratio (Fig. 3E; the area surrounded by a dotted or solid rectangle) indicate that αActinin in these areas is under greater tension, reflecting temporary tip and basal adhesion, respectively[23]. A heatmap of the relaxation ratio also revealed that the increase in tension at the tip and base of the filopodia was temporary and decreased over time (Fig. 3F).

Next, to investigate tension on αCatenin, we performed time-lapse superresolution imaging of NIH3T3 cells expressing the αCatenin tension indicator. As shown in Fig. 3G, the αCatenin tension indicator revealed a clear contrast in color tone with orange-red at the cell margins (dotted rectangle) and almost green at the cell protrusions (solid rectangle). This result indicates that there is more tension on αCatenin at cell margins and less tension at cell protrusions. Consistently, the cell margins did not move during time-lapse imaging, whereas the cell protrusions constantly deformed and moved (Fig. 3H, J, Supplementary Videos 3 and 4). We quantified the fluorescence intensity along the edge of the cell protrusions (Fig. 3H) and along the cell boundaries (Fig. 3J) and then calculated the relaxation ratio. The heatmap of the relaxation ratio of lamellipodia revealed a warmer color (Fig. 3I), whereas that of the cell margins presented a cooler color (Fig. 3K), when the same color scale was applied. Notably, both heatmaps exhibited a striped pattern, indicating that tension on αCatenin is relatively constant over time. These results revealed a sharp contrast between the relatively constant tension on αCatenin and the continuously fluctuating tension on αActinin.

## Tension heterogeneity in the Z-discs of cardiomyocytes in αActinin tension indicator mice

To visualize tension on αActinin in mice, we generated a gene-targeted mouse model. We knocked in the targeting vector that drives the expression of the αActinin tension indicator under the CAG promoter in a lineage-specific manner into the Hipp11 gene locus[24] (Fig. 4A, Supplementary Fig. 4A). We confirmed lineage-specific expression by crossing with cardiomyocyte-specific Troponin T-Cre mice [25] (Supplementary Fig. 4B). However, as the fluorescence was faint, hereafter, we used mice expressing the αActinin tension indicator throughout the body by crossing with Ayu-Cre mice[26,27]. We confirmed that αActinin-mCherry was properly localized to sarcomeres and adherens junctions in the heart by immunostaining with anti-sarcomeric (SA)-αActinin antibody and anti-α-catenin antibody and staining with phalloidin (Supplementary Fig. 4C) and that αActinin-mCherry interacts with endogenous αActinin in the heart (Supplementary Fig. 4D). In addition, the αActinin tension indicator mice presented normal cardiac function (Supplementary Fig. 4E). These results suggest that the αActinin tension indicator behaves in the same manner as the endogenous αActinin.

Next, we investigated αActinin tension in living organs. For this purpose, whole hearts and livers were harvested and examined by confocal fluorescence microscopy. Both αActinin-TS and αActinin-mCherry were easily visualized in both the heart and liver and were properly localized at the z-disc or cell-cell junction (Fig. 4B). Since the color tone was different along the z-disc in a mosaic manner in the heart (Fig. 4B, white rectangle), cardiomyocytes were isolated from αActinin tension indicator mice and examined via superresolution fluorescence microscopy. As expected, a mosaic pattern of color tone along the z-disc in cardiomyocytes was clearly observed (Fig. 4C). Quantitative analysis revealed that the relaxation ratio was indeed variable periodically along the z disc, indicating that tension on αActinin is variable along the z-disc in cardiomyocytes (Fig. 4C).

To exclude the possibility that this mosaic pattern arises from intermolecular FRET between the TS module and mCherry rather than genuine tension differences, we quantified FRET efficiency in MDCKII cells in vitro and cardiomyocytes in vivo expressing the αActinin tension indicator (Supplementary Fig. 4F, G). A weak FRET signal was detected only under conditions of artificially high transient overexpression in MDCKII cells, whereas no detectable FRET was observed at physiological expression levels (Supplementary Fig. 4F). Cardiomyocytes from αActinin tension indicator mice likewise exhibited no measurable FRET (Supplementary Fig. 4G). These results demonstrate that intermolecular FRET is negligible under physiological and in vivo conditions.

To analyze how tension is applied to αActinin in cardiomyocytes, blebbistatin was added to cardiomyocytes isolated from αActinin tension indicator mice, followed by time-lapse superresolution imaging. As shown in Fig. 4D and Supplementary Video 5, after the addition of blebbistatin, the color tone of the z-discs of a cardiomyocyte was clearly changed from orange-red to yellow-green in the merged image, and the relaxation ratio transitioned from cool to warm in color. In the magnified pictures, the change in color tone can be more clearly observed (Fig. 4E and Supplementary Video 6). Quantitative analysis revealed that the relaxation ratio was significantly increased after the addition of blebbistatin (Fig. 4F). The intensity of the green fluorescence of αActinin-TS markedly increased after the addition of blebbistatin, while the intensity of the red fluorescence of αActinin-mCherry was constant. These results indicate that the αActinin tension indicator faithfully represents αActinin tension in vivo.

## Tension on αCatenin is different from that on αActinin in vivo

Next, to visualize tension on αCatenin in mice, we generated a gene-targeted mouse expressing the αCatenin tension indicator with the same approach as that used with the αActinin tension indicator (Fig. 5A, Supplementary Fig. 4A). Because the fluorescence of αCatenin TS was also faint, we used a mouse expressing the αCatenin tension indicator throughout the body by crossing with Ayu-Cre. We confirmed that αCatenin-mCherry was properly localized on adherens junctions both in the heart and in the liver by immunostaining with an anti-αCatenin antibody and an anti-N-Cadherin antibody and staining with phalloidin (Supplementary Fig. 5A–D) and that αCatenin tension indicator mice presented normal cardiac function (Supplementary Fig. 5E), suggesting that the overexpression of the αCatenin tension indicator does not have any harmful effects.

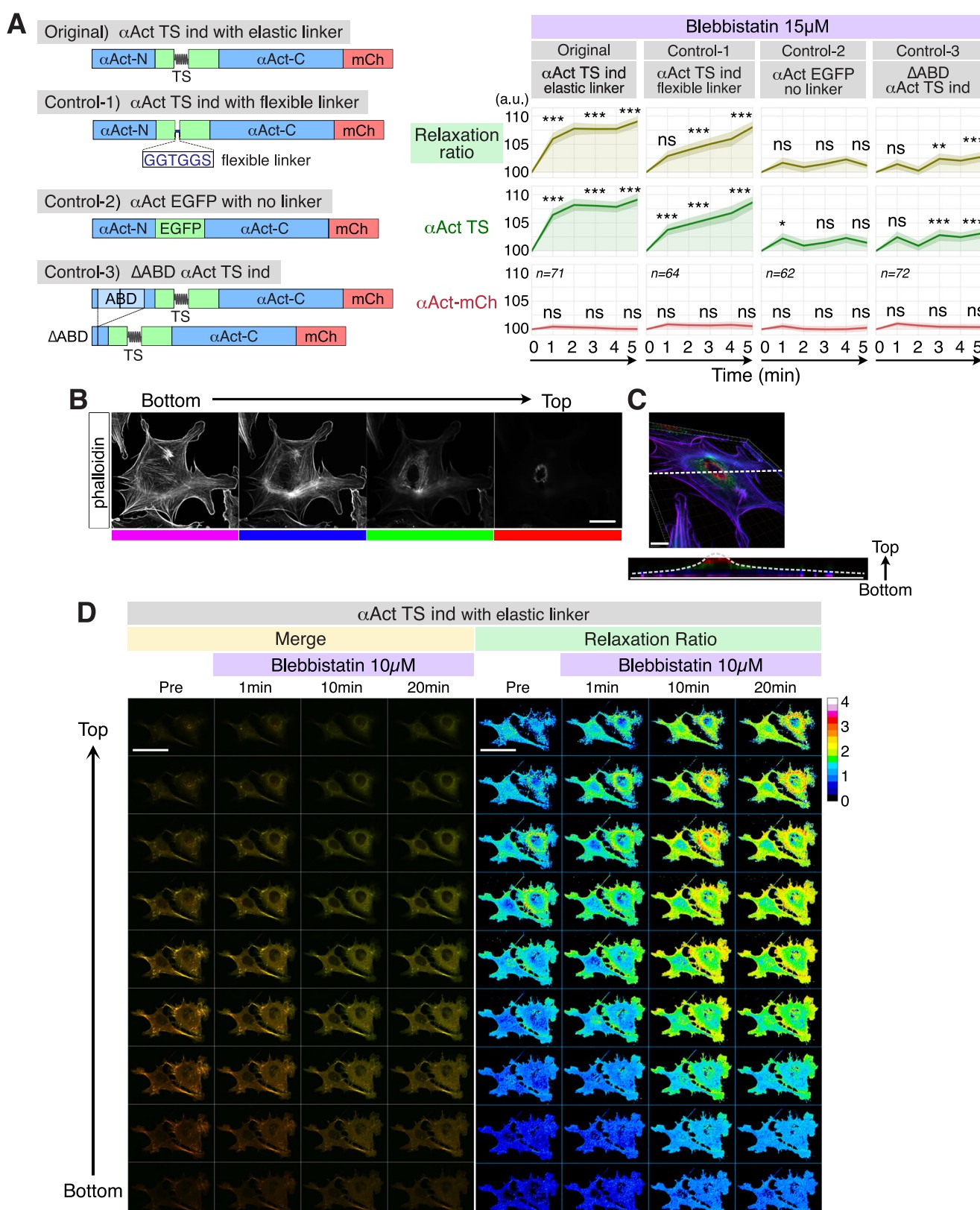

To examine tension on αCatenin in living organs, whole hearts and livers were harvested from αCatenin tension indicator mice subjected to confocal fluorescence imaging. As shown in Fig. 5B, both αCatenin-TS and αCatenin-mCherry were properly localized on cell-cell junctions in the heart and liver. In the heart, the color tone of the merged image of αCatenin-

TS and αCatenin-mCherry was orange to yellow-green with some variations [Fig. 5B, compare (i), (ii) and (iii)]. In contrast, in the liver, the color tone was almost red at most cell junctions, indicating that αCatenin was under enough tension that αCatenin-TS almost completely lost its green fluorescence (Fig. 5B). Interestingly, there were many bright yellow-green spots on

**Fig. 2 | Detection of intracellular tension on αActinin. A** Schematic illustration of the original αActinin tension indicator and its mutants, and quantification of time-lapse fluorescence microscopy of MDCKII cells expressing the original αActinin tension indicator and mutants before and after the addition of 15 μM blebbistatin. The shaded areas indicate the 95% CI. The exact sample size for each condition is shown directly on the plots ($n$ = number of cells). Statistical analysis was performed using a one-way ANOVA, followed by the Scheffé post hoc test; ns: $p > 0.05$, *: $p < 0.05$, **: $p < 0.01$, ***: $p < 0.001$. Asterisks indicate statistical significance at 1, 3, and 5 min compared to 0 min. **B** Z-stack confocal fluorescence images of an NIH3T3 cell stained with phalloidin. Note that the distribution of actin filaments differs depending on the z-section. Scale bar: 20 μm. **C** A three-dimensional reconstitution of the z-stack in (**B**) is shown. The color was coded according to the height of the z-stack, as indicated in (**B**). Scale bar: 20 μm. A cross-section along the dotted line shown in (**B**) is also displayed. **D** Time-lapse fluorescence microscopy of NIH3T3 cells expressing the αActinin tension indicator before and after the addition of 10 μM blebbistatin. Merged images of the green (αActinin-TS) and red (αActinin-mCherry) channels show height-dependent differences in color tone. Scale bar: 50 μm. Ratiometric images display pixel-wise green/red fluorescence (relaxation ratio) using a cool-warm colormap, where warm colors indicate lower tension and cool colors indicate higher tension. Scale bar: 50 μm.

the cell junctions (Fig. 5B, dotted circles). These spots indicate bile canaliculi surrounded by tight junctions (TJs), which were stained with an anti-ZO-1 antibody (Supplementary Fig. 5D). These results indicate that tension on αCatenin is more relaxed specifically around bile canaliculi, which are sealed mainly by tight junctions, whereas αCatenin is in tension at most cell-cell junctions in the liver (Fig. 5E).

### Tension on αCatenin is distinct from that on αActinin at cell junctions in liver

To compare tension on αCatenin with that on αActinin, the liver was harvested from each indicator mouse and was subjected to superresolution fluorescence imaging. As shown in Fig. 5C, the αCatenin tension indicator appeared green at tricellular junctions, which, similar to regions surrounding bile canaliculi, are sealed by tight junctions. In contrast, most bicellular junctions were red, indicating that αCatenin is under tension in the majority of junctional regions but remains relaxed at tricellular junctions where tight junction sealing is strongest. In contrast, the αActinin tension indicator showed a mosaic pattern along cell junctions (Fig. 5D), indicating that tension on αActinin is not constant and varies along cell junctions. These results indicate that tension on different molecules could differ even at the same junctional regions in the liver (Fig. 5E, F).

### Discussion

In this study, we developed αActinin and αCatenin tension indicators based on a circularly permuted EGFP module, enabling single-fluorophore visualization of intracellular forces while simultaneously reporting protein abundance through a C-terminal mCherry. These sensors allowed us to detect subtle, spatially heterogeneous tension patterns at high resolution in cultured cells and in mouse tissues, revealing unexpectedly complex and molecule-dependent force landscapes in vivo.

A striking example emerged in the liver, where the αCatenin tension indicator showed a markedly different pattern from the αActinin tension indicator. This difference can be rationalized by their distinct molecular roles: αCatenin directly couples cadherins to actin filaments and is continuously loaded by contractile forces at adherens junctions, whereas αActinin crosslinks separate actin filaments and can be either tensed or relaxed depending on filament geometry. In the liver, αCatenin tension was uniformly high along cell-cell adherens junctions but was markedly reduced around bile canaliculi. Consistent with this, our imaging also revealed that αCatenin is relaxed at tricellular junctions, apical interfaces that are sealed by tight junctions. These observations suggest that apical junctional structures divert mechanical load away from adherens junctions. This idea aligns with the known apical organization of hepatocytes, where tight junctions together with pericanalicular actomyosin rings form the primary load-bearing structure, thereby reducing the tensile load transmitted to adherens junctions at these interfaces. In contrast, the mosaic tension pattern along αActinin likely reflects local differences in the organization and orientation of actin filaments, which cause αActinin to experience varying mechanical loads even within the same junction. Together, such organization may help distribute forces efficiently and prevent structural damage under fluctuating mechanical loads, and further studies will be necessary to elucidate how these molecular and architectural features together shape tension transmission in vivo. Notably, these liver-specific patterns contrast with the dynamic, cyclic tension loaded on αActinin in cardiomyocytes,

underscoring how tissue-specific cytoskeletal architectures impose fundamentally different mechanical states in vivo.

Beyond enabling force visualization in vivo, the indicators provide several practical advantages for high-resolution imaging. The use of a single fluorophore allows tension to be inferred from fluorescence changes without requiring donor-acceptor FRET imaging enabling superresolution imaging potentially single-molecule visualization. Compared with previously reported FRET-based αActinin TSs[8], our single-fluorophore indicator provides a clear advantage in visualizing fine spatial variations in tension across lamellipodia, filopodia, and different z-planes. Despite its lower intrinsic brightness (Supplementary Fig. 2B), the cpEGFP-based TS provided sufficient signal-to-noise for high-resolution tension imaging in cells and tissues.

The sensitivity of our TS module was much higher when images were acquired with sCMOS (Zyla 4.2Plus®) or EMCCD (iXon®) detectors in the Dragonfly® confocal system than with PMT detectors in an FV3000® confocal microscope. Thus, a wide dynamic range is critical for detecting fluorescence changes, particularly when the fluorescence intensity varies across intracellular regions. We also revealed a quantitative correlation between force and fluorescence intensity of single-molecule TSs generated by circular permutation, which will be valuable for further sensor development. Although some GFP-based constructs have been reported to exhibit loading rate-dependent unfolding in single-molecule studies[28,29], our optical tweezer measurements revealed that the TS module undergoes rapid and reversible conformational changes under the loading rate used in our calibration (8–10 pN/s, corresponding to a pulling speed of 100 nm/s). The force-extension curves showed no detectable hysteresis over repeated pulling-relaxation cycles (Supplementary Fig. 1F), indicating that the fluorescence-relevant conformational changes are reversible and occur on a timescale faster than our loading protocol. While these properties support the robustness of the TS module as a force reporter, the fluorescence intensity of our TS was weaker than EGFP, and its dynamic range became narrow above 2 pN (Fig. 1K). Improving fluorescence brightness and optimizing the linker will broaden the detectable force range[30].

In the αCatenin tension indicator, we removed the domains that interact with vinculin to enhance fluorescence brightness. Because the M-domain–associated vinculin-binding site contributes to junctional reinforcement under tension[17], the indicator primarily reports α-catenin functions independent of vinculin-mediated stabilization. Endogenous αCatenin can still recruit vinculin and maintain the load-bearing complex, enabling reliable tension readout under physiological expression levels. However, when the indicator exceeds endogenous αCatenin protein levels, reduced vinculin reinforcement may lead to modest underestimation of local tension, which should be considered when interpreting the data. In summary, we developed tension indicators based on cpEGFP with improved sensitivity, resolution, and versatility, revealing unexpectedly complex and molecule-dependent tension landscapes in mouse tissues. These indicators offer powerful tools for dissecting mechanical forces in physiological, pathological, and developmental processes in vivo.

### Methods

#### Mice

All animal experiments were approved by the Animal Care and Use Committee of Kansai Medical University (approval number:

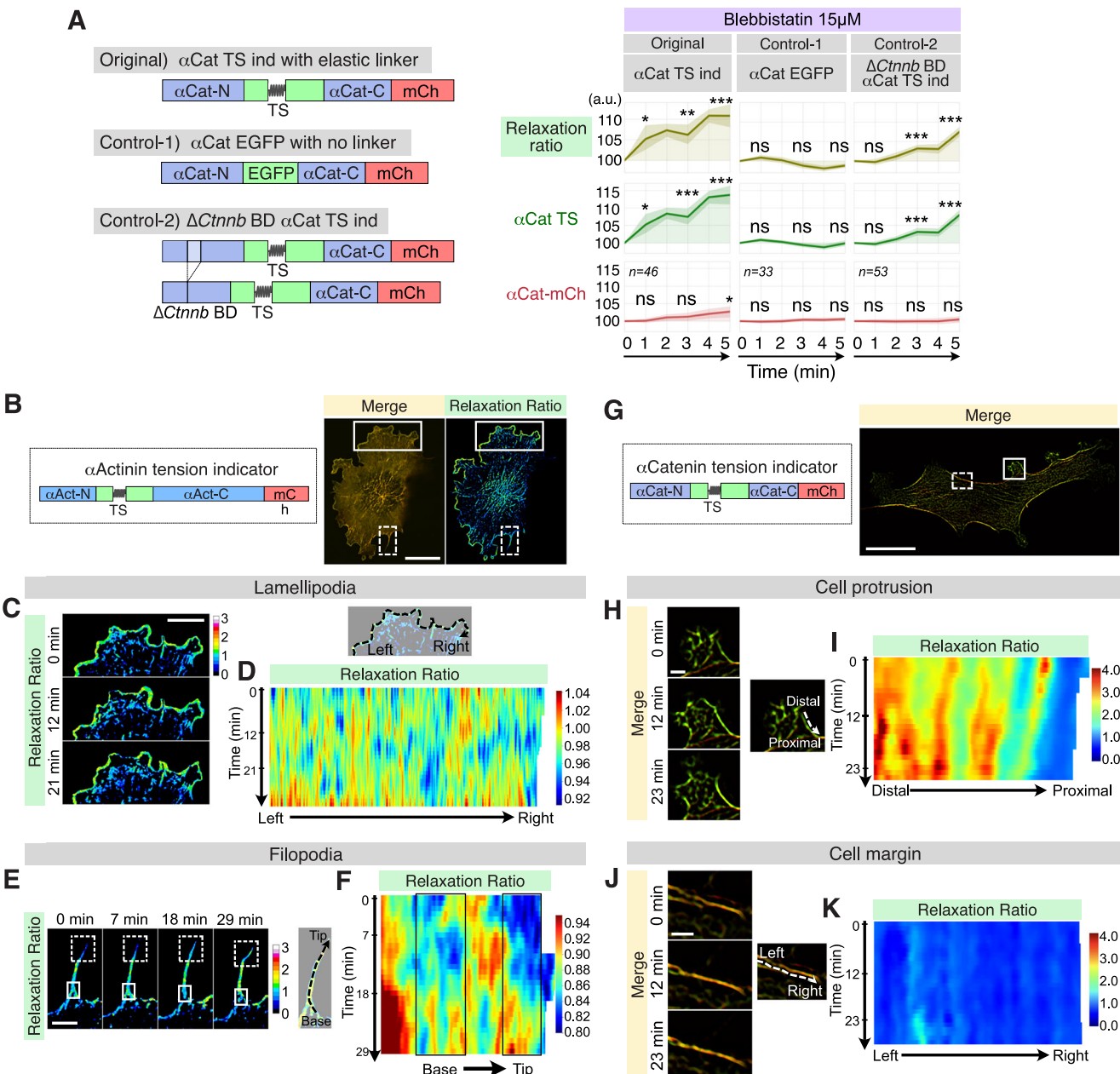

**Fig. 3 | Detection of tension dynamics with the αActinin and αCatenin tension indicators. A** Schematic illustration of the original αCatenin TS indicator and its control constructs, and quantification of time-lapse fluorescence microscopy of NIH3T3 cells expressing each construct before and after the addition of 15 μM blebbistatin. The shaded areas indicate the 95% CI. The exact sample size for each condition is shown directly on the plots ($n$ = number of cells). Statistical analysis was performed using one-way ANOVA followed by the Scheffé post hoc test; ns: $p > 0.05$, *: $p < 0.05$, **: $p < 0.01$, ***: $p < 0.001$. Asterisks indicate statistical significance at 1, 3, and 5 min compared to 0 min. **B** Domain structure of the αActinin tension indicator and a representative confocal fluorescence image of an NIH3T3 cell expressing the αActinin tension indicator. A merged image of αActinin-TS and αActinin-mCherry and a ratiometric image of the relaxation ratio are displayed. Scale bar: 20 μm. **C** Time-lapse fluorescence microscopy of lamellipodia. Magnified images correspond to the area surrounded by the solid rectangle in (**B**). The relaxation ratio shows warm colors in lamellipodia. Scale bar: 10 μm. **D** Heatmap of the relaxation ratio along the leading edge. The X-axis indicates position along the lamellipodia (left to right). The Y-axis represents time. Tension fluctuates

continuously without any regularity. **E** Time-lapse fluorescence microscopy of filopodia. Magnified images correspond to the area surrounded by the dotted rectangle in (**B**). Images of the relaxation ratio show that cool colors at the filopodial tip (dotted rectangles) and base (solid rectangles) indicate higher tension. Scale bar: 4 μm. **F** Heatmap of the relaxation ratio along the filopodia. The X-axis represents the position from base to tip. The Y-axis represents time. Higher tension is applied at the tip and base of the filopodia (rectangle), suggesting adhesion at the tip and base. **G** Domain structure of the αCatenin tension indicator, and a representative superresolution fluorescence image of an NIH3T3 cell expressing the αCatenin-tension indicator. Scale bar: 50 μm. **H** Time-lapse images of the cell protrusion area surrounded by the solid rectangle in (**G**). Scale bar: 5 μm. **I** Heatmap of the relaxation ratio along the dotted arrow on the cell protrusion. The X-axis indicates position (distal to proximal). The Y-axis indicates time. **J** Time-lapse images of the cell margin area surrounded by the dotted rectangle in (**G**). Scale bar: 5 μm. **K** Heatmap of the relaxation ratio along the dotted arrow on the cell margin. The X-axis indicates position (left to right). The Y-axis indicates time. Note the clear contrast between the heatmaps in (**I, K**).

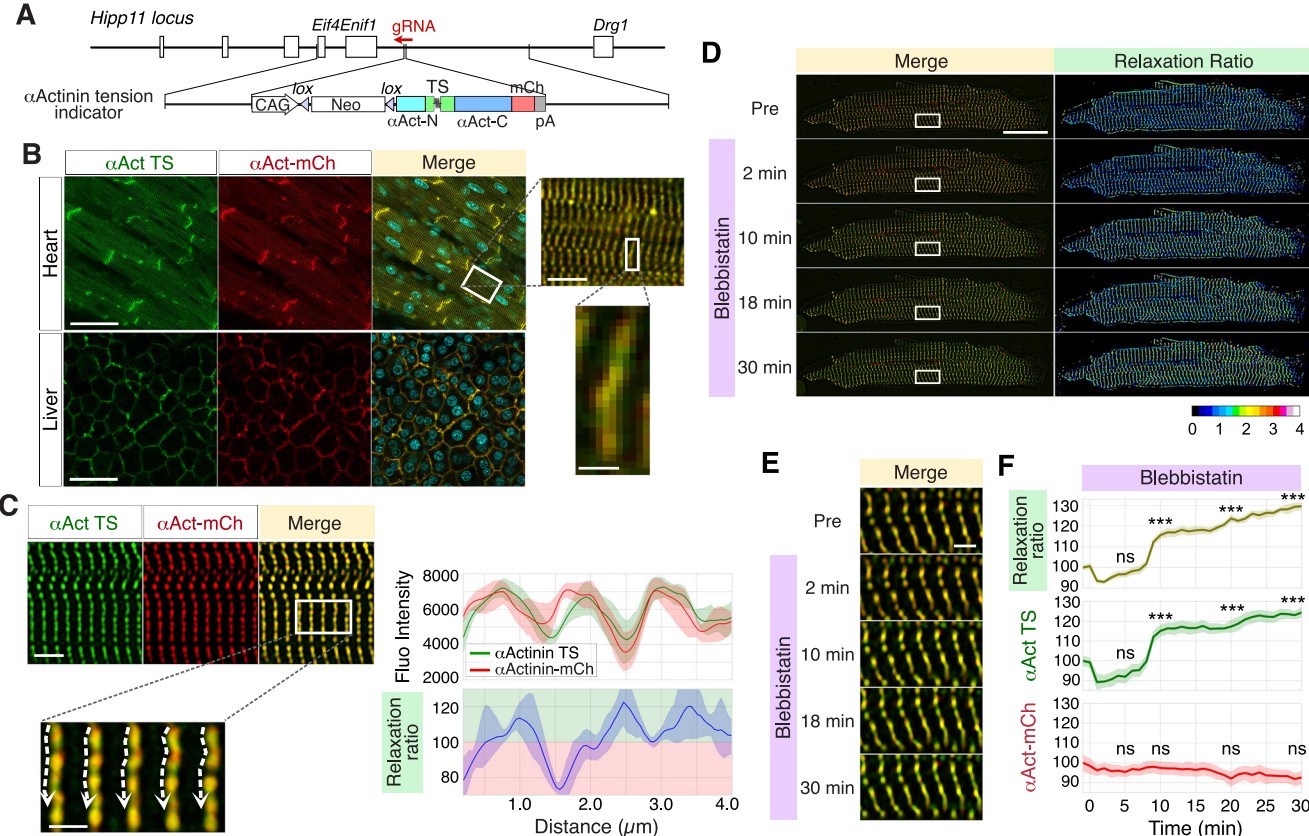

**Fig. 4 | Tension heterogeneity in the Z-discs of cardiomyocytes in αActinin tension indicator mice. A** The strategy for targeting the Hipp11 gene locus with the αActinin tension indicator. A floxed neomycin cassette and an αActinin tension indicator cassette were placed immediately after the CAG promoter. The αActinin tension indicator is expressed specifically in the cells that have expressed Cre. **B** Confocal fluorescence microscope images of the heart and liver of αActinin tension indicator mice labeled with Hoechst. Scale bar: 50 μm. A magnified image of the area surrounded by the rectangle shows that the fluorescence of αActinin tension indicator exhibits a striated pattern, consistent with the z-disc, where endogenous αActinin is localized. Scale bar: 10 μm. Another magnified image of the area surrounded by the rectangle shows that the merged fluorescence of αActinin-TS and αActinin-mCherry is variable along the z-disc, suggesting heterogeneity of the tension forces. Scale bar: 2 μm. **C** Superresolution fluorescence microscopy images of an isolated cardiomyocyte. Scale bar: 5 μm. A magnified image of the area surrounded by the rectangle is displayed. Scale bar: 2 μm. Quantitation of fluorescence intensity along the dotted lines and the relaxation ratio are shown. Shaded areas

indicate 95% confidence intervals (CIs) ($n$ = 5 z-discs). Variations in tension were observed along the dotted lines. **D** Time-lapse superresolution fluorescence microscopy images of a cardiomyocyte before and after the addition of blebbistatin. Merged images of αActinin TS and αActinin-mCherry, and images of the relaxation ratio are shown. Scale bar: 20 μm. Note that the relaxation ratio color shifted toward warm over time after the addition of blebbistatin. **E** Time-lapse magnified images of the region surrounded by the rectangle in (**D**). Note that the color of the merged image of αActinin-TS and αActinin-mCherry shifted toward green over time after the addition of blebbistatin. Scale bar: 2 μm. **F** Quantitation of the mean fluorescence intensity of αActinin-TS, αActinin-mCherry, and the relaxation ratio over time. The intensity of αActinin-TS markedly increased after the addition of blebbistatin, whereas the intensity of αActinin-mCherry remained constant. Shaded areas indicate 95% confidence intervals (CIs). Data represent $n$ = 10 z-discs from 2 independent experiments. Statistical analysis was performed using a one-way ANOVA followed by the Scheffé post hoc test; ns: $p > 0.05$, ***: $p < 0.001$. Asterisks indicate statistical significance compared to 0 min.

A2025-011). We have complied with all relevant ethical regulations for animal use. Outbred Slc:ICR mice (MGI:2175911; Japan SLC, Inc.) were used for wild-type controls and for the maintenance of the Cre lines and the gene-targeted mice generated in this study. The αActinin and αCatenin tension indicator lines were generated in this study as described in the "Generation of gene-targeted mice" section and was maintained on a Slc:ICR background. The Ayu-Cre line (Tg(Ayu1−cre)1Kyam; MGI:3688277), kindly provided by Dr. K Yamamura (Kumamoto University)[26], was maintained on a mixed ICR background. The Troponin T-Cre (TnT-Cre) line (B6.Cg−Tg(Tnnt2−cre)1Jsh/J; MGI:3714578) was purchased from The Jackson Laboratory (strain #: 024240)[25] and backcrossed with Slc:ICR mice. Male and female mice (4–6 weeks old) were used in equal numbers. All animals were maintained in a specific pathogen-free facility under a 12-h light/dark cycle with free access to food and water. For line maintenance and experimental crosses, Cre-positive mice were bred with Slc:ICR mice. Genotyping was performed by PCR using genomic DNA extracted from tail tips.

## Plasmids

pN1-G-CaMP8 was kindly provided by Dr. M Ohkura (Kyushu University of Health and Welfare)[16]. ActTS-GR, Sfp pet29b C-terminal His Tag, and pET28a-ybbR-His-sfGFP-DocI were purchased from Addgene (cat. 79774, 61563, 75015, and 58708), and pTYB21 vector was purchased from NEB (cat. N6709S). To generate TS (TS-2 in Fig. 1B), the coding sequence (CDS) of mutant cpEGFP within G-CAMP8 (nt 799…1521 of pN1-G-CaMP8) was cloned into pCR-Blunt II-TOPO vector (ThermoFisher, cat. 451245). The flexible linker GGTGGS of G-CaMP8 was replaced with an elastic linker $(GPGGA)_8$ which was cloned from ActTS-GR. Mutant cpEGFP and an elastic linker were amplified by PCR with either KOD Fx Neo (Toyobo, cat. KFX-201) or PrimeSTAR HS DNA Polymerase (Takara, cat. R010A), then assembled or ligated into pCR-Blunt II-TOPO vector (pCRII) with either In-Fusion HD Cloning Kit (Clontech, cat. 639649) or DNA Ligation Kit (Takara, cat. 6023). To generate another type of TS module (TS-1), N-terminal domain of EGFP (nt 1…435; nt 679…1113, U55762.1) was fused to the elastic linker and C-terminal domain of EGFP (nt 451…720; nt 1129…1398, U55762.1).

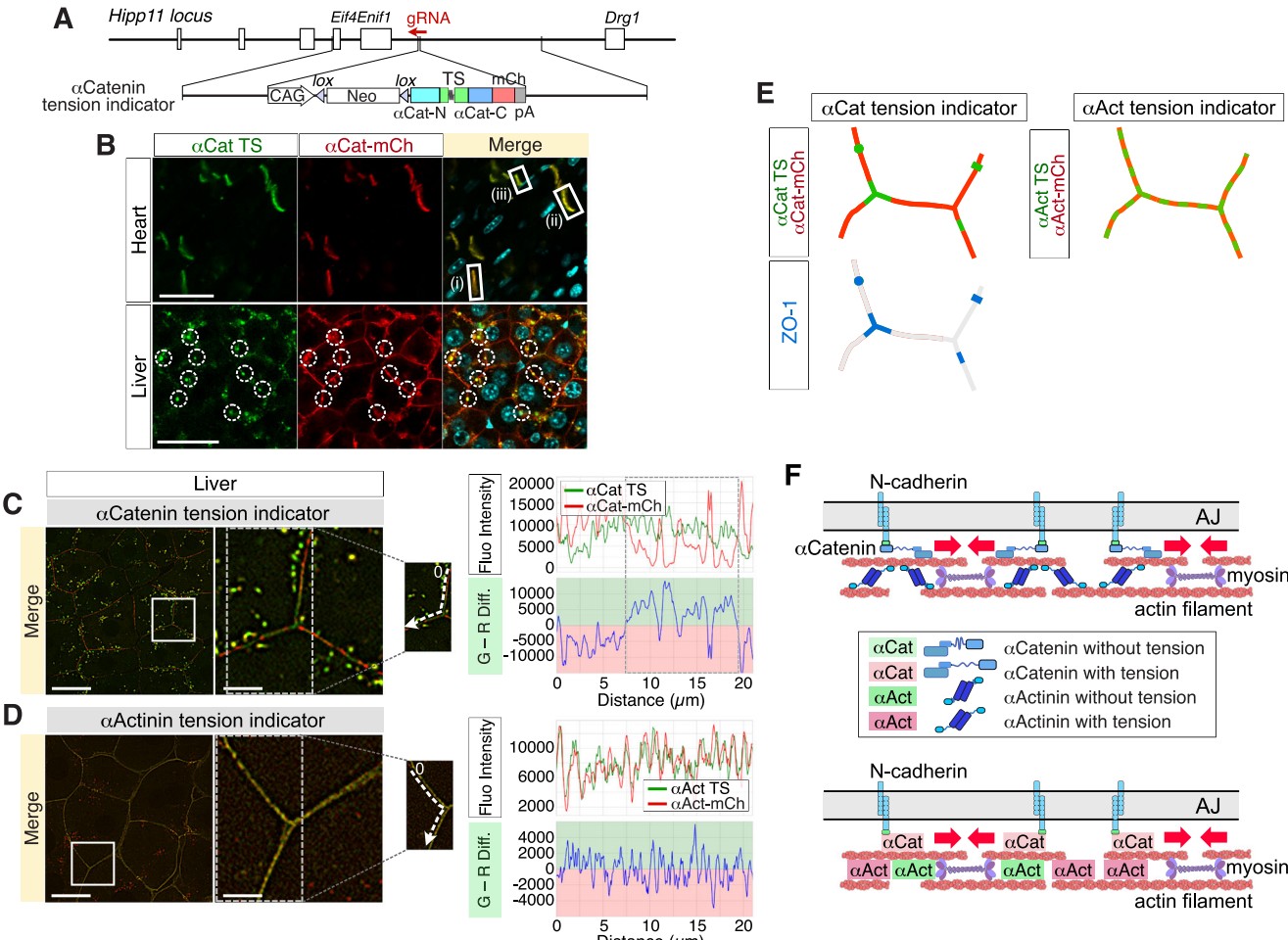

**Fig. 5 | Tension on αCatenin is distinct from that on αActinin at cell junctions in vivo. A** The strategy of Hipp11 gene locus-targeting of the αCatenin tension indicator (the same strategy used for generating αActinin tension indicator mouse). The αCatenin tension indicator is expressed specifically in the cells that have expressed Cre. **B** Confocal fluorescence microscopy images of the heart and liver of an αCatenin tension indicator mouse labeled with Hoechst. Scale bar: 40 μm. Note that the color tone is variable among cell junctions in the heart [white rectangles; (i-iii)], and yellow-green in the vicinity of bile canaliculi in the liver (dotted circles). **C, D** Superresolution fluorescence microscopy images of livers expressing either the αCatenin tension indicator or the αActinin tension indicator. Scale bar: 20 μm. Magnified images of the regions surrounded by the rectangles are shown. Scale bar: 5 μm. The merged image of the αCatenin tension indicator is red along most cell junctions and green in a subset of regions, presumably in the vicinity of bile canaliculi (**C**). In contrast, the merged image of the αActinin tension indicator shows a mosaic pattern of green and red along the junctions in (**D**). Quantitative data of fluorescence intensities and relaxation indices of αActinin and αCatenin tension indicators are displayed alongside the images. The fluorescence intensities at the cell junctions were measured along the dotted arrows. The dotted rectangle indicates the region where the green fluorescence of the αCatenin TS is dominant over the red fluorescence of αCatenin-mCherry. The relaxation index was calculated by subtracting the red fluorescence intensity of αCatenin-mCherry from the green fluorescence intensity of the αCatenin TS. These plots are intended for within-sensor qualitative comparison of spatial patterns and are not used for cross-sensor quantitative comparison. **E** Schematic illustration of cell junctions in the liver with αActinin and αCatenin tension indicators. **F** Schematic illustration of how the difference between tension on αActinin and that on αCatenin is caused at the same loci.

To generate αActinin tension indicator, ActTS-GR containing full-length *Actn1* CDS (nt 247…2922, NM_001102.4) was utilized. After AgeI/NotI digestion, either TS-1 or TS-2 described above was amplified and inserted in *Actn1* (between nt 1210 and 1211, NM_001102.4). Subsequently, mCherry was fused in-frame at the 3' end right before the stop codon of *Actn1* CDS.

For cloning of *Ctnna1* total cDNA was synthesized with RNA purified from mouse kidney using SuperScript III First-Strand Synthesis System with random hexamers (ThermoFisher, cat. 18080051). Fragments of *Ctnna1* CDS was amplified from total cDNA by PCR, assembled into pBluescript II SK(+) with In-Fusion HD Cloning Kit. Thus, full-length *Ctnna1* CDS of nt 70–2790 of mouse catenin (cadherin associated protein), alpha 1 (NM_009818.1; Ctnna1) was cloned. The EcoRI and MfeI sites were introduced between nt 1017/1018, and between nt 1989/1990, respectively. To generate αCatenin tension indicator, *Ctnna1* CDS between nt 1018/1989 was replaced with TS module (TS-2) connecting with flexible linkers at both

ends (5'-side: GGSGGGSG, 3'-side: GSGCGS, notated as amino acids (aa)). The 3' end of this modified *Ctnna1* was fused to mCherry.

After NotI site of pCRII vector was disrupted by self-ligation after NotI/PvuI digestion, pCRII vector was modified by inserting "CAG promoter – floxed neomycin-resistance gene – multiple cloning sites (MCS) cassette". The 5' and 3' arm of targeting vector for H11 genomic locus are nt 3193195–3195267 and nt 3195268–3198203 of NC_000077.7 respectively, which were amplified from genomic DNA purified from kidney of C57BL/6J mouse by PCR. These 5' and 3' arm were assembled into the modified pCRII vector described above to generate H11 targeting vector. Either αActinin or αCatenin tension indicator cassette was inserted into the MCS site of this H11 targeting vector.

For bacterial expression of TS, pET28a-ybbR-His-Flag-TS-GST-ybbR vector was generated. The TS with Flag tag at the N-terminus and GST and ybbR sequence at the C-terminus was amplified and inserted into pET28a-ybbR-His-sfGFP-DocI (Addgene, cat. 58708) after NheI/XhoI digestion.

## Cell lines and culture

MDCKII cells (ECACC, cat. 00062107) and NIH3T3 cells (ATCC, cat. CRL-1658) were cultured at 37 °C and 5% $CO_2$ in Dulbecco's Modified Eagle Medium (DMEM) (ThermoFisher, cat. 11965092) supplemented with 10% FBS (vol/vol) (Biosera, cat. FB-1365/500) and Penicillin-Streptomycin-L-Glutamine (ThermoFisher, 100x, 1:100, cat. 10378016). Cells at ~80% confluency on ø3.5-cm plate were transfected with the expression vector of each TS indicator using Lipofectamine 2000 Transfection Reagent (ThermoFisher, cat. 11668019). After overnight incubation, 40,000 cells in 200 µl of culture medium were plated onto the glass-bottom area of ø3.5-cm of glass-bottom dish (Matsunami Glass, cat. D11131H) which had been treated with 0.1% gelatin (vol/vol) (Sigma Aldrich, cat. G1890-100G). The next day, 2.0 mL of culture medium was added, followed by fluorescence imaging.

## Generation of gene-targeted mice

EGR-05 ES cells derived from B6/129 F1 mice were kindly provided by Dr. M. Ikawa (Osaka University). Mouse embryonic fibroblasts (MEFs) were purchased (Merck Millipore cat. PMEF-NL-C), thawed onto a ø15-cm dish in DMEM supplemented with 10% FBS (vol/vol) and Penicillin-Streptomycin-L-Glutamine, expanded onto four ø15-cm dishes, then treated with Mitomycin C (MMC) (Kyowa-Kirin, 1 mg/mL, 1:100) at 37 °C, 5% $CO_2$ for 2 h. MEFs were stored in 90% FBS/10% Dimethyl Sulfoxide (DMSO) (Sigma-Aldrich, cat. D2650) in liquid $N_2$. ES cells were maintained on MMC-treated MEFs in advance in ES medium consisting of DMEM supplemented with 5% FBS (vol/vol), 16% KnockOut™ Serum Replacement (vol/vol) (ThermoFisher, cat. 10828028), EmbryoMax Nucleoside (Merck Millipore, 100x, 1:100, cat. ES-008-D), MEM Non-Essential Amino Acids Solution (ThermoFisher, 100x, 1:100, cat. 11140050), Sodium Pyruvate (ThermoFisher, 100 mM, 1:100, cat. 11360070), 2-Mercaptoethanol (2-ME) (ThermoFisher, 1:1000, cat. 21985023), ESGRO Recombinant Mouse LIF Protein (Sigma-Aldrich, $10^7$ units/mL, 1:6000, cat. ESG1107), and Penicillin-Streptomycin-L-Glutamine. For gene-targeting, 10,000 ES cells were plated onto each well of a 6-well plate where MEFs had been seeded. The next day, ES cells were transfected with 1.2 µg of targeting vector (the full DNA sequences for "pCR_H11_CAG_-floxNeo_αActTS_indicator_targeting_vector" and "pCR_H11_CAG_-floxNeo_αCatTS_indicator_targeting_vector" are provided in Supplementary Data 2), 1.0 µg of pX330-H11-gRNA, and 0.1 µg of pEF1-Puro using Lipofectamine LTX with Plus Reagent (ThermoFisher, cat. 15338500). For pX330-H11-gRNA, the following oligonucleotides for a gRNA targeting the H11 gene locus were synthesized by Hokkaido System Science, annealed, and cloned into BbsI site of pX330-gRNA vector (Addgene plasmid #158973):

Sense: 5'-caccATGATGGCATCTAATGAGCT-3',
Anti-sense: 5'-aaacAGCTCATTAGATGCCATCAT-3'

The gRNA was designed using the CRISPR Design tool (formerly crispr.mit.edu; Zhang Lab, MIT)[31], and validated with CRISPOR. This analysis yielded a high MIT specificity score of 78, confirming the absence of off-target sites with fewer than three mismatches in protein-coding exons. Given this high specificity and the subsequent backcrossing to segregate potential off-target mutations, experimental off-target assessment was not performed. After selection with 1 µg/mL puromycin dihydrochloride (ThermoFisher, cat. A1113803) and 200 µg/mL G418 Sulfate (ThermoFisher, Geneticin™ Selective Antibiotic, cat. 10131027) for 48 h, ES cells were cultured without puromycin for an additional 2 days. Then, ES cells were replated at a density of 400 cells/well onto MEF-seeded 6-well plates and cultured with 200 µg/mL G418 for 7 days. Eight clones of ES cells from each line were picked up, trypsinized, and plated onto a 96-well plate where MEFs had been seeded and cultured for additional 3 days. After splitting into three 96-well plates, two were stored at -80 °C, and another one was used for genotyping. Proper homologous recombination at both 5'- and 3'- ends was confirmed by PCR using KOD Fx Neo (Toyobo, cat. KFX-201) to exclude random integration.          PCR was carried out with an initial

denaturation at 94 °C for 2 min, followed by 32 cycles of 98 °C for 10 s, 58 °C for 30 s, and 68 °C for 2.5 min. PCR products were separated on a 1% agarose gel stained with ethidium bromide and visualized using a UV transilluminator. Primers were designed to span the homology arms (one primer located outside the targeting vector and the other within the insert sequence). The primer sequences were as follows:

5'-junction_fwd: 5'-tcaccacatccacctccttgtaagtatc-3'
5'-junction_rev: 5'-aggaaagtcccataaggtcatgtactgg-3'
3'-junction_fwd: 5'-actacaccatcgtggaacagtacgaacg-3'
3'-junction_rev: 5'-ctggttttctcactaggacattgactgatagg-3'

After confirmation, ES cells from a positive clone were injected into 8-cell-stage Slc:ICR embryos (Japan SLC). The next day, blastocysts were transplanted into the uteri of pseudo-pregnant ICR mice. After chimeric mice were born, the desired gene-targeted mice were obtained through germ-line transmission. A single F0 founder was used to establish each line. The lines were maintained by backcrossing with Slc:ICR mice (Japan SLC). Genotyping was performed with PCR using KOD Fx Neo (Toyobo, cat. KFX-201) with three primers below, yielding 428 bp (wild-type) and 283 bp (targeted) amplicons:

H11_p1_fwd: 5'-aatccttcagctgcccactctactg-3'
H11_p2_rev: 5'-tcaggacctctgaaagaccagcta-3'
H11_p3_rev: 5'-cctattggcgttactatgggaaca-3'

The mouse line expressing either αActinin or αCatenin tension indicator throughout the body was produced by crossing with Ayu-Cre mice[27].

## Purification of TS and Sfp protein and oligo-arm attachment

For bacterial expression of TS, the pET28a-ybbR-His-Flag-TS-GST-ybbR vector was generated. The N-terminal Flag-tagged TS and C-terminal GST and ybbR sequences were amplified and inserted into pET28a-ybbR-His-sfGFP-DocI (Addgene, cat. 58708) after NheI/XhoI digestion. BL21 bacteria were transformed with the pET28a-ybbR-His-Flag-TSmod-GST-ybbR vector, expanded with 200 mL LB culture medium in the presence of kanamycin (50 µg/mL) (Wako, cat. 113-00343), followed by induction with isopropyl-β-D(-)-thiogalactopyranoside (IPTG: final 0.5 mM) (Wako, cat. 096-05143) at 32 °C for 3 h. Bacteria were collected by centrifugation at 4 °C for 5 min at $10,000 \times g$, washed once with TE buffer (10 mM Tris-HCl, pH 8, 1 mM EDTA), and resuspended in 10 mL PBS supplemented with lysozyme (1 mg/mL) (Wako, cat. 123-06721) and protease inhibitor cocktail (final: 1x, Nacalai, cat. 04080). Resuspended bacteria were sonicated in ice-cold water for 30 min (repeated cycle of 30 s sonication – 10 sec pause) with ultrasonic processor (Astrason cat. XL2020). After centrifugation at 4 °C for 15 min at $10,000 \times g$, the supernatant was filtered through 0.45 µm filter, then mixed with 6 mL bed volume (BV) of Ni-NTA agarose (Qiagen, cat. 30210), and incubated with rotation at 4 °C for 1 h. After centrifugation at 4 °C for 2 min at 3000 rpm, the agarose mixture was applied to four poly-prep chromatography columns (1.5 mL BV each) (Bio-Rad, cat. 731-1550), each column was washed with 5 mL PBS twice, 5 mL 10 mM imidazole (Wako, cat. 099-00015), and 1.2 mL 200 mM imidazole, then eluted with 3 mL of 200 mM imidazole. The eluted TS protein (total 12 mL) was dialyzed in PBS at 4 °C overnight. Subsequently, 1.5 mL BV of glutathione agarose (Thermo Scientific, cat. 16100) was applied to a poly-prep chromatography column. After washing with 10 mL PBS, the eluted TS protein was applied to this glutathione agarose column, washed with 5 mL PBS and 1.2 mL of elution buffer (20 mM reduced glutathione (Wako, cat. 077-02013), 100 mM Tris-HCl, pH 8, and 120 mM NaCl), then eluted with 3 mL of elution buffer. Purified TS protein was concentrated with Amicon Ultra-0.5 10 kDa MWCO centrifugal filter (Millipore, cat. UFC501024) and stored at −80 °C.

Conjugation of CoA-modified oligos to TS protein followed the protocol of protein labeling and tethering kit (ybbR) (Lumicks)[32]. Briefly, 10 µmol of purified TS protein was incubated with Sfp enzyme, 3'-CoA modified oligo, and TCEP in 1x Sfp reaction buffer at RT for 3 h. TS-oligo chimeric protein was purified with His Spintrap (Cytiva, cat. 28-4013-53), concentrated with Amicon Ultra-0.5 10 kDa MWCO centrifugal filter, and stored at −80 °C.

### Generation of DNA handles and C-trap

DNA handles (5 kb) were PCR-amplified from the pTYB21 vector (NEB, cat. N6709S) using Taq polymerase (Takara, cat. R001A) with a uracil-containing forward primer and a biotinylated reverse primer, treated with USER enzyme (NEB, cat. M5505S), annealed to a ligation oligo, ligated with T4 ligase (NEB, cat. M0202S), and purified by ExoI digestion and agarose gel extraction. TS proteins carrying complementary oligo arms were assembled with the DNA handles and tethered between two streptavidin beads (Spherotech) trapped in the C-trap optical tweezers (Lumicks) in imaging buffer (40 mM Tris-HCl pH 7.5, 150 mM NaCl, 7.5 mM $MgCl_2$, 1 mM DTT, 0.5 mg/ml BSA, 2 mM Trolox, 10 nM catalase, 37.5 μM glucose oxidase, 30 mM glucose). One optical trap held one bead stationary, while the other was displaced at a constant velocity of 100 nm/s to apply mechanical tension. Simultaneously, fluorescence from the TS module was recorded under 488-nm excitation. Force-extension curves were obtained by converting bead-to-bead distances to molecular extension after subtracting the theoretical response of the 10-kb dsDNA handles using the Marko–Siggia worm-like chain (WLC) model $x(F) = L\left(1, |, \sqrt{k_B T/(4FP)}\right)$ with $L = 3.4\text{Å} \times 10{,}000$, $P = 50$ nm, and $k_B T = 4.1$ pN·nm. Protein extension was defined relative to a 1.25-pN reference point as $x_{prot} = \Delta d_{meas} - \Delta x_{DNA}$. Because this raw extension systematically overestimated the mechanical response of the TS, curves were normalized using the known elasticity of the 40-aa flexible linker (GPGGA)$_8$ (contour length ≈15.2 nm), whose WLC prediction at 8 pN is $x_{linker}(8\,\text{pN}) \approx 6.6$ nm; a single multiplicative factor was applied such that $x_{prot}(8\,\text{pN}) = 6.6$ nm for all traces. Upper-envelope curves were extracted by binning protein extension and taking the maximal force per bin, and were subsequently fitted using locally weighted regression (LOESS) to obtain a smooth force–extension mean curve. The variability (shading) was computed by LOESS-smoothing the absolute residuals from the mean, yielding a position-dependent SD band.

5k handle forward primer: 5'-ccUUccggcUggcUggUUUaUUgctg-3'
5k handle reverse primer: (BIO) – 5'-ttaagctagcttacttgtacagctcg-3'
ssDNA to ligate: 5'- gttgggacgggtcaccgcctaccaatagcaacaacgccttccgg ctggctggtttatt-3'

Quantification of TS fluorescence was performed using Fiji, and C-trap datasets were visualized using Lakeview (Lumicks). Raw HDF5 files were imported into Python (3.12.2) using the h5py library for further analysis. Plots were generated using pandas (2.2.3), NumPy (1.26.4), SciPy (1.13.1), Matplotlib (3.3.4), Seaborn (0.13.2), and scikit-learn (1.5.2). Fluorescence traces were smoothed using exponentially weighted averaging (pandas), and curve fitting was carried out using SciPy's least_squares. Fitting errors were quantified using the mean_squared_error metric from scikit-learn.

### Heart dissociation and isolation of cardiomyocytes

Six-week-old mice were injected intraperitoneally with 150 μL of heparin (150IU; Mochida Pharmaceutical). Five minutes later, mice were euthanized by cervical dislocation, and the heart was harvested, and the ascending aorta was cannulated with a blunt-end 18 G needle, then perfused with 15 mL of perfusion buffer consisting of Hanks' Balanced Salt Solution (HBSS) (Gibco, 10x, 1:10, cat. 14185052) supplemented with 0.02% MgSO4 (Wako, cat. 131-00405), 0.06% taurine (Wako, cat. 201-00112), 0.08% glucose (Sigma-Aldrich, cat. 07-0680), and 0.1% 2,3-butanedione monoxime (BDM) (Sigma-Aldrich, cat. B0753-25G). Cannulated heart was subsequently perfused with collagenase type2 (Worthington, cat. CLS-2) in perfusion buffer at 37 °C using a Langendorff apparatus. After gentle pipetting with large-bore glass pipette in perfusion buffer containing 4% bovine serum albumin (BSA) (Wako, cat. 013-27054), dissociated cardiomyocytes were plated on a ø3.5-cm glass-bottom dish (Matsunami, cat. D11131H) treated with laminin (1:100, Sigma-Aldrich, cat. L2020-1MG). Attached cardiomyocytes were subjected to immunostaining or live imaging.

### Immunofluorescence

Six-week-old mice were intraperitoneally injected with 150 μL of heparin (150 IU, Mochida). Five minutes later, mice were euthanized by cervical

dislocation, and the heart was harvested. The ascending aorta was cannulated with a blunt-end 18G needle. Through this needle, the heart was perfused with 15 ml of 30 mM KCl (Nacalai, cat. 28514-75) in 1x phosphate buffered saline (PBS) (Wako, 10x, 1:10, cat. 048-29805), and then subsequently perfused with 5 ml of 4% paraformaldehyde (PFA) in PBS (Wako, cat. 163-20145). After additional fixation in 4%PFA at 4 °C overnight, the heart was soaked in 20% sucrose in PBS at 4 °C overnight, then embedded in Tissue-Tek OCT compound (Sakura, cat. 4583). In the case with anti-αCatenin rabbit polyclonal antibody, either the heart or liver was harvested without fixation, soaked in 20% sucrose in PBS at 4 °C for 30 min, and then embedded in OCT. In this case, frozen sections were fixed with acetone for 5 min at −20 °C, then subjected to antibody staining.

For immunostaining of mouse tissue, frozen sections were rinsed with PBS. After permeabilization with 0.1% Triton (Nacalai, cat. 35501-15) in PBS ("PBSTr", hereafter) at room temperature (RT) for 10 min, sections were subjected to blocking with 10% donkey serum, 3% skim milk, AffiniPure Fab Fragment Goat Anti-Mouse IgG (H + L) (1:100, Jackson ImmunoResearch, cat. 115-007-003) in PBSTr at RT for 1 h. After rinse with PBSTr twice, sections were incubated with primary antibodies at 4 °C overnight. After washing in PBSTr at RT for 10 min for three times, sections were incubated with secondary antibodies, Hoechst 33342 (1 mg/mL, 1:100), with or without phalloidin, at RT for 90 min. After washing in PBSTr at RT for 10 min three times, sections were mounted with fluorescence mounting medium (Agilent Dako, cat S3023). Primary and secondary antibodies were diluted in 10% donkey serum, 3% skim milk in PBSTr prior to incubation with frozen sections. Primary antibodies used; anti-αActinin (Sarcomeric) mouse monoclonal (1:200, clone EA53, Sigma-Aldrich, cat. A7732); anti-αCatenin rabbit polyclonal (1:200, Sigma-Aldrich, cat. C2081); anti-N-Cadherin mouse monoclonal (1:200, clone 32, BD Bioscience, cat. 610920); anti-ZO-1 rabbit polyclonal (1:200, ThermoFischer, cat 40-2200). Secondary antibodies used were Alexa Fluor 488, 546, or 647, anti-rabbit, mouse, or rat IgG (ThermoFisher). Phalloidin staining was performed with Alexa Fluor 647 Phalloidin (ThermoFisher, cat A222287) or phalloidin-iFluor 488 Conjugate (1:1000, Cayman Chemical, cat 20549). Nuclear staining was performed with Hoechst 33342 (1 mg/mL, 1:50).

### Fluorescence imaging

Fluorescence imaging was performed on three types of confocal microscope systems depending on experimental purposes.

For multipoint time-lapse imaging, confocal microscope C1 (Nikon) mounted on a Nikon Eclipse Ti base equipped with a Perfect Focus System (PFS), a PlanApo 20x/0.75 NA objective, a PlanApo 40x/0.95 NA objective was used. A solid-state 405-nm laser at 30 mW (CVI Melles Griot 405), solid-state Coherent Sapphire 488-nm laser at 18.4 mW, and solid-state 561-nm laser at 10.8 mW (CVI Melles Griot 561) were used. Images were acquired using EZ-C1 software. The 405-nm laser and 450/35-nm filter were used for Hoechst staining, 488-nm laser and 525/50-nm filter for TS, and 561-nm laser and 595/40-nm filter for mCherry.

For imaging of immunostaining samples, FV3000 confocal laser scanning microscope (Olympus) equipped with a High Sensitivity-Spectral Detector (HSD), Spectral Detector (SD), a TruFocus Z-Drift Compensation Module (ZDC), an UPLSAPO40X2/0.95 NA objective, a PLAPON60XOSC2 60x/1.4 NA oil immersion objective, and an UPLSAPO100XO/1.4 NA oil immersion objective was used. A solid-state 405-nm (50 mW), solid-state 488-nm (20 mW), solid-state 561-nm (20 mW), solid-state 640-nm (40 mW) diode lasers together with the main laser combiner package (FV31-MCOMB-P) were used. Images were acquired with FLUOVIEW FV3000 software (Olympus). The 561- and 647-nm channels were detected with HSD, while 405- and 488-nm channels were detected with SD.

For time-lapse imaging with simultaneous acquisition of two colors or superresolution imaging, Dragonfly 500 spinning disk confocal system (Andor Technology) on an Olympus IX83 base equipped with a Zyla4.2 Plus scientific complementary metal oxide semiconductor (sCMOS) image sensor and an iXon Life 888 electron multiplying charge-coupled device (EMCCD) image sensor, UPLXAPO20X/0.4 NA objective,

UPLXAPO60xO/1.42 NA oil immersion objective, or UPLXAPO 100XO/1.45 NA oil immersion objective was used. A solid-state 405-nm (100 mW), 488-nm (50 mW), 561-nm (50 mW), and 637-nm (140 mW) smart laser were used. Images were acquired with Fusion software (Andor Technology), Hoechst was excited at 405 nm and detected with 445/46-nm, GFP/TS/Alexa488 at 488 nm with 521/38-nm, mCherry/Alexa-555 at 561 nm with 594/43-nm, and Alexa-647 at 637 nm with 698/77-nm filters. For SRRF-Stream super-resolution imaging, 20 frames (live imaging) or 50 frames (immunostaining) were acquired using an iXon Life 888 EMCCD camera, followed by processing with Fusion software (Oxford instruments).

Images used for quantitation of Figs. 1F, 2A, 3A, Supplementary Figs. 1C, D, 2A–E, and 3A–C were taken on the Nikon C1 microscope, images in Figs. 1B, E, 4B, 5B, Supplementary Figs. 1B, 4E, F, and 5A–D were taken with the FV3000 microscope, and images in Figs. 2B–D, 3B–K, 4C–F, 5C, D, and Supplementary Fig. 4C were taken on the DragonFly imaging system.

## Acceptor-photobleaching FRET analysis

Intermolecular FRET between the GFP donor in the TS module and the mCherry acceptor was quantified using an acceptor-photobleaching FRET assay. MDCKII cells transiently expressing the αActinin TS indicator ($n = 25$) or cardiomyocytes isolated from αActinin TS indicator mice ($n = 8$), were plated on glass-bottom dishes coated with fibronectin (1:200; Sigma-Aldrich, cat. F1141-5MG) or laminin (1:100; Sigma-Aldrich, cat. L2020-1MG) and fixed with 4% PFA for 10 min. Fixed samples were imaged on an Olympus FV3000 confocal microscope using a UPLSAPO40x/0.95 NA objective with a 2× zoom. Three images were acquired before acceptor bleaching ("Pre"), after which mCherry was bleached using high-intensity 561-nm laser irradiation. Three additional images were then collected under identical settings ("Post"). GFP (donor) and mCherry (acceptor) fluorescence intensities were extracted from identical ROIs in the Pre and Post images. For each cell, the average intensity of the three Pre images ($G_{pre}$, $R_{pre}$) and the three Post images ($G_{post}$, $R_{post}$) was used for downstream calculations. Bleach efficiency, donor increase, and corrected FRET efficiency were calculated as follows:

$$\text{Bleach efficiency}\,(\%) = 100 \times \left(1 - \frac{R_{post}}{R_{pre}}\right)$$

$$\text{Donor increase}\,(\%) = 100 \times \left(\frac{G_{post}}{G_{pre}} - 1\right)$$

$$\text{Corrected FRET efficiency} = \frac{1 - (G_{pre}/G_{post})}{1 - (R_{post}/R_{pre})}$$

For statistics, mean ± SD were calculated for each parameter. A paired $t$-test was performed to compare donor fluorescence before and after bleaching ($G_{pre}$ vs. $G_{post}$).

## Protein extraction and Immunoprecipitation

Each frozen ventricular tissue sample from either the αActinin or αCatenin tension indicator mouse was crushed into pieces with a hammer and resuspend in 4 mL of ice-cold 1x PBS. After centrifugation at $500 \times g$ for 2 min at 4 °C, the supernatant was removed. The pellet was resuspended in 1.4 mL of lysis buffer (1x PBS, 1% Triton, 1x protease inhibitor (Nacalai, cat. 03969-21)), transferred to a tissue homogenizer (Sansho, cat. 81-0780), and homogenized with 16 strokes on ice. After centrifugation at 15,000 rpm for 5 min at 4 °C the supernatant was collected as the heart lysate. Protein concentration was measured using Bio-Rad Protein Assay Kit (Bio-Rad, cat. 5000001), and 100 μg of protein lysate was used for immunoprecipitation. Protein lysates were precleared by rotation with 15 μL of protein G agarose (bed volume: 7.5 μL) conjugated to normal rabbit IgG (Santa-Cruz, cat. sc2027) at 4 °C for 1 h, washed three times with 1 mL of 0.1% PBSTr, and then subjected to immunoprecipitation with 15 μL of anti-mCherry antibody (abcam, cat. ab167453) by rotation at 4 °C for overnight. After three washes with 1 mL of 0.1% PBSTr, immune complexes were eluted with 40 μL of glycine-HCl (pH3.2) at RT for 5 min. Following centrifugation, the supernatant was neutralized by adding 4 μL of 0.5 M Tris-HCl (pH 8), 1.5 M NaCl, 16 μL of 4x NuPAGE™LDS Sample Buffer (ThermoFisher, cat. NP0007), and 6.5 μL of 10x NuPAGE™ Sample Reducing Agent (ThermoFisher, cat. NP0009), then denatured by heating at 70 °C for 10 min. Samples were separated by SDS-PAGE on NuPAGE™ 4–12% Bis-Tris, 1.0-mm, Mini Protein Gel (4–12%) (ThermoFisher) in NuPAGE™ MOPS SDS Running Buffer (ThermoFisher), followed by Western blotting. Primary antibodies used were anti-mCherry polyclonal (1:1000, abcam, cat. ab167453), anti-αActinin polyclonal (Proteintech, cat. 11313-2-AP). Secondary antibody used were rabbit IgG HRP conjugated polyclonal (1:3000, R&D Systems, cat. HAF008) or mouse IgG HRP conjugated polyclonal (1:3000, R&D Systems, cat. HAF007). For chemiluminescence detection, Immobilon Forte Western HRP substrate (Millipore, cat. WBLUF0500) or ImmunoStar LD (Wako, cat. 292-69903) was used. Protein band imaging was performed with FUSION Solo S (Vilber Lourmat, France).

## Image analysis and quantification

All the images acquired with a microscope were processed with Fiji. 3D image reconstitution was done by Imaris 10.2 (Oxford instruments). Chromatic shifts in SRRF-Stream super-resolution images were corrected using Chromagon (Matsuda et al.[33]). For time-lapse confocal imaging (Nikon A1R), regions of interest (ROIs) were manually drawn around individual cells in Fiji, and the integrated fluorescence intensities ("IntDen") of the green (TS donor) and red (mCherry acceptor) channels were extracted for each time point. For each ROI and each channel, fluorescence values were normalized to the first acquired frame ($t = 0$) according to:

$$F(t) = 100 \times \frac{I(t)}{I(0)},$$

where $I(t)$ denotes the raw integrated intensity at time $t$ and $F(t)$ represents the normalized fluorescence expressed as a percentage of the initial intensity. This normalization was applied independently to the green and red channels to yield $F_G(t)$ and $F_R(t)$.

To quantify conformational changes in the TS construct, the green-to-red ratio was computed at each time point:

$$R_{GR}(t) = \frac{I_G(t)}{I_R(t)}$$

The ratio was further normalized to its value at $t = 0$ to obtain a relaxation ratio:

$$\text{Relaxation ratio}(t) = 100 \times \frac{R_{GR}(t)}{R_{GR}(0)}.$$

Relaxation ratio heatmaps (Figs. 2D, 3B–F, I, K, and 4D) were generated by calculating pixel-wise green/red fluorescence ratios and rendering them as pseudo-color images without any masking, intensity gating, or weighting based on mCherry signal. Color-scale ranges were adjusted per figure panel to optimize visibility: fixed color scales were used for within-cell comparisons (Fig. 3I, K), while narrower scales were applied for images with limited dynamic range (e.g., Fig. 3D, F).

For SRRF-Stream images acquired with the DragonFly spinning disk system, the same normalization was applied to ROIs on intracellular structures. These images were reconstructed using the SRRF algorithm, which enhances local contrast and sharpens spatial features. No additional filtering or masking was applied. Local green–red differences were calculated as:

$$\Delta_{GR}(t) = F_G(t) - F_R(t)$$

All quantitative analyses and plots were generated using Python 3.11.12 and its scientific ecosystem, including pandas 2.3.3, NumPy 1.26.4, SciPy 1.15.2, Matplotlib 3.10.3, and Seaborn 0.13.2. Schematic diagrams were prepared using Affinity Designer 2.5.7.

## Statistics and reproducibility

Statistical analyses were conducted using Python 3.11.12 with SciPy 1.15.2, pandas 2.3.3, and NumPy 1.26.4. Experiments were performed independently at least two or three times with similar results to ensure reproducibility. The specific number of independent replicates (n) and the definition of sample size for each experiment are provided in the corresponding figure legends. For comparisons between two independent groups, independent-samples *t*-tests were used. For paired measurements obtained from the same ROI or the same cell before and after treatment, Wilcoxon signed-rank tests were performed. When comparing more than two groups, one-way ANOVA was applied, followed by Scheffé's post-hoc test to identify statistically significant differences. Unless otherwise specified, data are presented as mean ±95% confidence interval, calculated from the standard error of the mean. $P < 0.05$ were considered statistically significant. The statistical test used for each experiment is noted in the corresponding figure legend.

## Reporting summary

Further information on research design is available in the Nature Portfolio Reporting Summary linked to this article.

## Data availability

The data supporting the findings of this study are available within the paper and its Supplementary Information. Plasmids generated in this study have been deposited in Addgene under accession codes: 252385 and 252386. Uncropped and unedited images of blots and gels are available in the Supplementary Information. Numerical source data for graphs and charts can be found in Supplementary Data 1. All other data are available from the corresponding author upon reasonable request.

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

## Acknowledgements

We thank N. Watanabe (Kyoto University, Kyoto, Japan), S.H. Yoshimura (Kyoto University, Kyoto, Japan), Y. Kamioka (Kansai Medical University, Osaka, Japan) and S. Hirano (Kansai Medical University, Osaka, Japan), for insightful discussions, M. Ohkura (Kyushu University of Health and Welfare, Miyazaki, Japan) for providing an expression plasmid of G-CaMP8. This project was supported by grants from Japan Society for the Promotion of Science (16H07353, 17K09586, 20K08502, and 23K07589 to M.H.), TAKEDA Science Foundation, The Novartis Foundation (16-113), SENSHIN Medical Research Foundation to M.H, and the Strategic Project for Proofreading and Submission Support of International Academic Papers by Kansai Medical University to M.H. This research was also supported by Research Support Project for Life Science and Drug Discovery (Basis for Supporting Innovative Drug Discovery and Life Science Research (BINDS)) from AMED under Grant Number JP24ama121020.

## Author contributions

M.H. conceived the study, developed the methodology, and supervised the project. K. Fujiwara, K. Fujiki, and M.H. performed the investigation. K. Fujiki, K.S., I.S., T.N., and M.H. provided resources. M.H. performed formal analysis and visualization, and wrote, edited, and finalized the manuscript. T.O.A. and T.N. contributed to the writing (review and editing).

## Competing interests

The authors declare no competing interests.
