## [Transparent Peer Review file · Communications Biology]

Molecular tension indicators reveal unexpectedly complex regulation of tension in live mouse organs

Corresponding Author: Dr Maretoshi Hirai

Version 0:

Reviewer comments:

Reviewer #1

(Remarks to the Author)

This is a very interesting manuscript by Fujiwara et al, in which the authors develop force biosensors that are dependent on fluorescence. Although a number of FRET-based force sensors have been developed, this work is innovative for constructing a fluorescent-based approach where an elastic peptide is inserted within eGFP. As force increases the GFP loses fluorescence. An mcherry fluorescent protein (at the end of the protein) allows for ratiometric imaging since the mcherry fluorescence is unaffected by force.

I really liked this approach for designing biosensors for measuring force as it allows for wider adoption of these biosensors (there is no need for FRET microscopy). This alone merits publication.

The authors validate this biosensor using blebbistatin as well as optical trap (to do a force calibration).

They also show that these sensors work in vivo, another strength of this paper and further validating the impact of this technique.

I have a few points I would like the authors to address:

1. Why is alpha-catenin studied in NIH3T3? In Figure 3F they reference protrusions, but in the earlier figures they show alpha-catenin at cell-cell adhesions. I felt this sensor should have been instead validated in a cell line that has well formed cell-cell adhesions, such as MDCK.
2. The talin sensor in Supplemental figure 1C isn't validated/quantified. This should be done if the authors wish to show an additional functional sensor.
3. The finding of lower tension at some regions in the bile canaliculi is interesting but not discussed well, leaving the reader to wonder what does this finding mean in terms of liver physiology? How does this fit into the existing literature on the biophysical state of these cells? Beyond this I felt the animal work is interesting as a proof of principle but the authors fail to help the reader understand the significance of their findings of force between tissues (heart vs liver) or with sub-cellular structures of the cell. Can the authors provide additional discussion and help connect their findings to existing literature?
4. The first paragraph in the discussion felt very out of place. Did this paragraph belong at the end of the results or end of the discussion?

Reviewer #2

(Remarks to the Author)

This study reports a single-fluorophore, non-FRET molecular tension indicator—an important technical advance with clear potential to address limitations of FRET sensors in vivo. By fusing the sensor to an mCherry internal reference, the authors enable simultaneous readout of tension and total protein. Impressively, they apply these reporters in transgenic mice and explore α -actinin and α -catenin tension patterns in intact organs, leading to the intriguing idea that in vivo mechanoregulation can be molecule-specific. The concept is novel, technically challenging, and the observations are thought-provoking. Before publication, however, the following points should be addressed:

1. Although drug perturbations and single-molecule calibration show the sensor is responsive to mechanical input, true reversibility remains unproven. It is unclear whether the probe can undergo multiple cycles of force application and recovery without performance loss—this hinges on whether the cpEGFP module can fully re-fold after force-induced conformational changes. Clarifying this (e.g., by washout or load-unload analyses) would strengthen the claims.

2. cpEGFP-based probes are likely influenced by intracellular pH, Cl^- , refractive index, and viscosity. In addition, while

composite pseudocolor aids visualization, please state whether spectral bleed-through corrections were applied.

3. The α -catenin reporter removes the M-region that mediates vinculin binding. Could this alteration affect the physiological tension experienced by endogenous α -catenin?

4. Because the force-to-fluorescence mapping may differ between α -actinin and α -catenin reporters, direct comparison of color scales/ratios across sensors is potentially unreliable. Please frame Figure 5 as within-sensor (qualitative) comparisons and avoid cross-sensor quantitative interpretations.

5. The parameters for the optical tweezer calibration are not sufficiently described. Specifically, the loading rate is a critical parameter that strongly influences protein unfolding forces. To ensure physiological relevance, the loading rate should mimic that experienced in the cellular environment, which recent work suggests is in the range of ~ 1 pN/s (see Hu et al., Cell, 187, 3445, (2024)). The authors should report their loading rate and discuss how it compares to the physiological regime, as this impacts the interpretation of the sensor's working force range.

Reviewer #3

(Remarks to the Author)

SUMMARY:

Fujiwara et al present a novel type of molecular tension sensor that does not rely on Förster Resonance Energy Transfer (FRET) as many tension sensors do, making it more accessible for certain applications. The authors take advantage of a circularly permuted version of EGFP (cpEGFP) that displays a range of intensities dependent on the force exerted on the molecule containing the cpEGFP. This allows for measurement of forces in the range of 0-6 pN using a single fluorescent protein. Fujiwara et al test this novel tension sensor design within the known force sensitive molecules α -actinin and α -catenin both in cells in culture and in mouse tissues. The authors also tag the molecules of interest with a red fluorescent mCherry protein at the C-terminus to track sensor localization and concentration. The sensors can apparently detect a range of forces in high spatiotemporal resolution, offering a promising approach for force sensing that requires less cumbersome analysis and could be more amenable to in vivo use.

Overall, this work could be a good proof-of-concept of an alternative force sensing module. I have some concerns as to how the sensors are portrayed relative to state-of-the-art FRET sensors, the experimental techniques proving functionality, and the unclear processing of image data. These concerns must be addressed to be confident in the functionality of these new sensors.

MAJOR COMMENTS:

- Some of the statements made about FRET-based tension sensors are factually incorrect. I agree that this proposed alternative has some advantages, but the state-of-the-art needs to be portrayed accurately. Here are the topics that should be addressed:

o While the original vinculin tension sensor was generated with a spider silk (GPGGA)_n linker, there have been many other linkers used that result in different extension properties of the sensors, including increases in the dynamic range [<https://pmc.ncbi.nlm.nih.gov/articles/PMC6053308/>].

o While intermolecular FRET can theoretically be an issue, it can and should be quantified by simultaneously expressing constructs with individual fluorophores so that the risk can be tested before drawing conclusions [<https://pubmed.ncbi.nlm.nih.gov/30523675/>]. In the absence of intermolecular FRET, differences in tension sensor expression should have no impact on FRET signal.

o The amount of FRET TS proteins is not uncertain. The concentration of the FRET sensor is measured by imaging the acceptor fluorophore alone with its excitation and emission wavelengths. Fluorescence of the acceptor is not dependent on energy transfer, and this approach has been used in most FRET TS papers to measure sensor concentration, including those cited by the authors.

o FRET is not "simply based on the distance between two independent fluorescent proteins", it is also dependent on relative orientation of the fluorescent protein dipoles. This may actually help the argument for the use of cpEGFP if the authors are confident that changes in fluorescence of cpEGFP are only due to distance between the GFP domains.

o FRET-based tension sensors have been successfully used in vivo. I have listed here only a few examples. [<https://pmc.ncbi.nlm.nih.gov/articles/PMC11874908/>, <https://www.nature.com/articles/s41598-023-50142-z>, <https://journals.plos.org/plosbiology/article?id=10.1371/journal.pbio.3000057>].

- The authors make heavy use of the myosin II inhibitor blebbistatin to demonstrate the functionality of their sensors. Blebbistatin is blue-light sensitive and becomes phototoxic upon stimulation with the wavelength of light required to excite EGFP. Additionally, blebbistatin is highly fluorescent in aqueous solutions greater than 10 μ M. It is not used in live cell imaging for these reasons. There are alternatives that have been recently developed, such as para-aminoblebbistatin [<https://pmc.ncbi.nlm.nih.gov/articles/PMC4886532/>]. If the reagent used in this work is a different form of blebbistatin, this should be clearly stated in the methods. If the authors used regular blebbistatin, I hesitate to draw conclusions from any experiments involving the application of blebbistatin to live cells and imaged using blue light.

- Fluorescent proteins are easily photobleached under typical imaging conditions, especially when repeatedly imaged over

time. Different fluorescent proteins bleach at different rates, which can make it difficult to measure FRET during a time series. It is imperative to show that photobleaching is not affecting the timelapse measurements. While it is promising that the untreated alpha-actinin TS and mCherry fluorescence stays stable in Figure 1f, there seems to be a large drop in fluorescence in the first time points in Figure 4f. One way to accomplish this would be to perform an experiment on fixed samples expressing the tension sensors with the same imaging parameters used in the various timelapse experiments and show that relaxation ratio is unaffected by repeated imaging.

- Is the data in Figure 1f, S1d, 2a, etc being normalized in some way to start at 100 a.u.? Is this obscuring variability in the starting fluorescence levels? This is especially relevant for comparing the different controls in 2a – I would assume Control-3 would start out at a higher EGFP intensity than the original sensor. It is important to be able to use a tension sensor to read out differences in tension levels across groups, not just in a single group in response to a treatment.

MINOR COMMENTS:

- FRET can occur between EGFP and mCherry [<https://www.nature.com/articles/s41598-023-50142-z>, <https://journals.plos.org/plosone/article?id=10.1371/journal.pone.0001916>]. As alpha-actinin functions as an antiparallel dimer, it is possible that intermolecular FRET occurs between the EGFP of one molecule and the mCherry of another. It would be best practice to perform an intermolecular FRET control by co-expressing alpha-actinin-cpEGFP and alpha-actinin-mCherry.

- Alpha-actinin more specifically localizes to actin filaments, not just “the cytosol” as stated in line 20.

- The talin cpEGFP tension sensor was disregarded due to the weak fluorescence, but is that not indicative of high levels of tension? Does signal get restored with blebbistatin (or other cytoskeletal inhibitor) treatment? I would be interested in following up on this finding. However, if this sensor is really not going to be explored further in this manuscript, I would leave it out entirely.

- Please explain what you mean by “the fluorescence of the alpha-actinin tension indicator changed from orange to yellow-green”. It needs to be clear that you are talking about an overlay/merge (I assume) of the mCherry and EGFP signals. Please also explain the “relaxation ratio” in bit more detail and how it can be interpreted.

- I don't understand in Figure 3d how both the relaxation ratio and the tension ratio range from 0-3 if they are just inverse of each other. Is there something else going on in this calculation? Since this is the only place that tension ratio is used, I would suggest leaving it out and just inverting the color map for the relaxation ratio. Unless something more complicated is going on that is not clearly explained.

- Is the data in Figure 3f from a single fibroblast not in contact with other cells? In this scenario, how is alpha-catenin under any tension without the existence of adherens junctions?

Version 1:

Reviewer comments:

Reviewer #2

(Remarks to the Author)

The authors have satisfactorily addressed all the concerns raised in my previous review. I find that the revisions have clarified the disputed points and strengthened the overall conclusions. I have no further comments and recommend the publication of the manuscript in its current form.

Reviewer #3

(Remarks to the Author)

The authors have satisfactorily addressed my concerns with the manuscript. I appreciate the additional experiments. Although they likely took some time to accomplish, they greatly strengthen the interpretation of the data. I am very excited about these tools and will explore using them in the near future.

We would like to thank the Reviewers for their careful evaluation and for providing insightful and constructive feedback. By addressing these comments, we feel that we have significantly improved the quality of our manuscript. Alterations to the manuscript in response to Reviewer's comments have been shaded in gray, to facilitate detection by Reviewers.

Reviewers' comments:

Reviewer #1 (Remarks to the Author):

This is a very interesting manuscript by Fujiwara et al, in which the authors develop force biosensors that are dependent on fluorescence. Although a number of FRET-based force sensors have been developed, this work is innovative for constructing a fluorescent-based approach where an elastic peptide is inserted within eGFP. As force increases the GFP loses fluorescence. An mcherry fluorescent protein (at the end of the protein) allows for ratiometric imaging since the mcherry fluorescence is unaffected by force. I really liked this approach for designing biosensors for measuring force as it allows for wider adoption of these biosensors (there is no need for FRET microscopy). This alone merits publication. The authors validate this biosensor using blebbistatin as well as optical trap (to do a force calibration). They also show that these sensors work in vivo, another strength of this paper and further validating the impact of this technique.

We appreciate the reviewer's positive assessment of the overall approach and validation strategy. The encouraging comments regarding the design, calibration, and in vivo applicability of our tension sensors are greatly valued. We address the specific concerns point-by-point below.

I have a few points I would like the authors to address:

1. Why is alpha-catenin studied in NIH3T3? In Figure 3F they reference protrusions, but in the earlier figures they show alpha-catenin at cell-cell adhesions. I felt this sensor should have been instead validated in a cell line that has well formed cell-cell adhesions, such as MDCK.

Thank you for this comment. We agree that MDCK cells would be a suitable system for our α Catenin tension indicator. We therefore attempted to validate the indicator in MDCKII cells; however, the fluorescence signal was too dim for reliable analysis because the α Catenin TS is intrinsically dimmer and MDCKII cells exhibit low transfection efficiency, resulting in insufficient signal for quantitative measurements. For this reason, we used NIH3T3 cells, in which the indicator is more robustly expressed and clearly detectable.

2. The talin sensor in Supplemental figure 1C isn't validated/quantified. This should be done if the authors wish to show an additional functional sensor.

We appreciate the reviewer's interest in the talin tension indicator. However, the fluorescence of this construct was too dim to allow reliable validation, and we did not generate a mouse line for it. In line with the suggestion from another reviewer, we have therefore removed the talin sensor from the manuscript to avoid presenting an incompletely validated sensor. Instead, we now provide extensive quantitative validation of the α Catenin tension indicator using appropriate control constructs. These results are presented in Figure 3A and Supplementary Fig. 3A-C and are described in the Results section (page 7, lines 14-26).

3. The finding of lower tension at some regions in the bile canaliculi is interesting but not discussed well, leaving the reader to wonder what does this finding mean in terms of liver physiology? How does this fit into the existing literature on the biophysical state of these cells? Beyond this I felt the animal work is interesting as a proof of principle but the authors fail to help the reader understand the significance of their findings of force between tissues (heart vs liver) or with sub-cellular structures of the cell. Can the

authors provide additional discussion and help connect their findings to existing literature?

We thank the reviewer for this insightful comment. As suggested, we have expanded the Discussion to clarify the physiological significance of the reduced α Catenin tension near bile canaliculi and to better integrate our findings with the known apical-lateral mechanical organization of hepatocytes. In the revised manuscript, we added the following sentences to explain why α Catenin tension is specifically relaxed around bile canaliculi (page 11, lines 12-19):

“In the liver, α Catenin tension was uniformly high along cell-cell adherens junctions but was markedly reduced around bile canaliculi. Consistent with this, our imaging also revealed that α Catenin is relaxed at tricellular junctions, apical interfaces that are sealed by tight junctions. These observations suggest that apical junctional structures divert mechanical load away from adherens junctions. This idea aligns with the known apical organization of hepatocytes, where tight junctions together with pericanalicular actomyosin ring form the primary load-bearing structure, thereby reducing the tensile load transmitted to adherens junctions at these interfaces.”

We also clarified the contrasting mosaic α Actinin tension pattern and its structural basis by adding (page 11, lines 19-21):

“In contrast, the mosaic tension pattern along α Actinin likely reflects local differences in the organization and orientation of actin filaments, which cause α Actinin to experience varying mechanical loads even within the same junction.”

Furthermore, to address the reviewer’s request to relate these observations to tissue-level mechanical contexts (heart vs. liver), we added (page 11, lines 24-26):

“Notably, these liver-specific patterns contrast with the dynamic, cyclic tension loaded on α Actinin in cardiomyocytes, underscoring how tissue-specific cytoskeletal architectures impose fundamentally different mechanical states in vivo.”

4. The first paragraph in the discussion felt very out of place. Did this paragraph belong at the end of the results or end of the discussion?

As suggested by the reviewer, we revised the Results and Discussion. Rather than beginning the Discussion with a paragraph that closely interprets Fig. 5, we incorporated the essential points of that paragraph into the final part of the Results so that the interpretation appears together with the corresponding schematic figure in Fig. 5E and F.

Reviewer #2 (Remarks to the Author):

This study reports a single-fluorophore, non-FRET molecular tension indicator—an important technical advance with clear potential to address limitations of FRET sensors in vivo. By fusing the sensor to an mCherry internal reference, the authors enable simultaneous readout of tension and total protein. Impressively, they apply these reporters in transgenic mice and explore α -actinin and α -catenin tension patterns in intact organs, leading to the intriguing idea that in vivo mechanoregulation can be molecule-specific. The concept is novel, technically challenging, and the observations are thought-provoking. Before publication, however, the following points should be addressed:

We sincerely appreciate the reviewer's thoughtful and generous assessment of our study. We are grateful for the recognition of both the technical advances and the biological insights offered by our single-fluorophore tension indicators. We have carefully addressed each of the points raised, and our detailed responses are provided below.

1. Although drug perturbations and single-molecule calibration show the sensor is responsive to mechanical input, true reversibility remains unproven. It is unclear whether the probe can undergo multiple cycles of force application and recovery without performance loss—this hinges on whether the cpEGFP module can fully re-fold after force-induced conformational changes. Clarifying this (e.g., by washout or load-unload analyses) would strengthen the claims.

We appreciate the reviewer's insightful comment. To directly address the concern about reversibility, we performed additional single-molecule optical tweezer experiments. As shown in the revised manuscript (Supplementary Fig. 1F), individual TS molecules exhibited nearly identical force-extension curves across three consecutive loading-unloading cycles, with no detectable hysteresis. These results demonstrate that the cpEGFP-based TS undergoes reversible conformational changes and can report multiple cycles of force application without performance loss. We have incorporated these new findings and clarified this point in both the Results (page 6, lines 8-10) and the Discussion (page 12, lines 8-14) sections of the revised manuscript.

2. cpEGFP-based probes are likely influenced by intracellular pH, Cl⁻, refractive index, and viscosity. In addition, while composite pseudocolor aids visualization, please state whether spectral bleed-through corrections were applied.

We appreciate the reviewer's insightful comments. Regarding spectral bleed-through, our confocal imaging was performed using sequential line scanning with 488-nm and 561-nm excitation, combined with low laser power, low PMT voltage, and no digital gain. In cells expressing the α -actinin tension indicator (which contains both the TS and mCherry), excitation with 488 nm did not produce any detectable signal in the red channel, and excitation with 561 nm did not produce any detectable signal in the green channel. Any cross-talk was not noticeable under our acquisition settings; therefore, no additional bleed-through correction was applied.

With respect to environmental sensitivity, the TS fluorescence was stable between pH 6–8 and showed detectable changes only under non-physiological extremes (increased green fluorescence at pH 11 and decreased fluorescence at pH 3). Thus, pH variation within living cells is unlikely to affect our measurements. We did not directly examine the effects of chloride concentration, refractive index, or viscosity; however, these parameters are expected to vary only modestly within the intracellular environments where α -Actinin and α -Catenin are localized. Therefore, their influence on TS is likely minimal under our experimental conditions.

3. The α -catenin reporter removes the M-region that mediates vinculin binding. Could this alteration affect the physiological tension experienced by endogenous α -catenin?

We thank the reviewer for highlighting this important point. We recognize this concern as well, as deletion of the α Catenin M-domain removes the vinculin-binding site (VBS), which is normally exposed under tension (PMID: 20453849). As a result, our M-domain deleted α Cat tension indicator cannot be stabilized by vinculin under tension.

However, when endogenous α Catenin is present, those molecules remain capable of recruiting vinculin and reinforcing the load-bearing structure. Under these physiological conditions, our indicator should still accurately report local tension. Nevertheless, we acknowledge that when our indicator is overexpressed relative to endogenous α Catenin, the absence of vinculin-mediated reinforcement may lead to a modest underestimation of tension. We have added the following clarification to the revised Discussion (page 12, lines 18-25):

“In the α Catenin tension indicator, we removed the domains that interact with vinculin to enhance fluorescence brightness. Because the M-domain-associated vinculin-binding site contributes to junctional reinforcement under tension (Yonemura et al., 2010), the indicator primarily reports α -catenin functions independent of vinculin-mediated stabilization. Endogenous α Catenin can still recruit vinculin and maintain the load-bearing complex, enabling reliable tension readout under physiological expression levels. However, when the indicator exceeds endogenous α Catenin protein levels, reduced vinculin reinforcement may lead to modest underestimation of local tension, which should be considered when interpreting the data.”

4. Because the force-to-fluorescence mapping may differ between α -actinin and α -catenin reporters, direct comparison of color scales/ratios across sensors is potentially unreliable. Please frame Figure 5 as within-sensor (qualitative) comparisons and avoid cross-sensor quantitative interpretations.

We appreciate the reviewer's thoughtful comment. We fully agree that the force-to-fluorescence relationship may differ between the α Actinin and α Catenin tension indicators, and that direct quantitative comparison across sensors should be avoided. Our intention in Figure 5 was to present within-sensor, qualitative comparisons of spatial tension patterns, rather than to infer quantitative differences between the two indicators. To prevent any potential misinterpretation, we have revised the figure legend as follows (page 29, lines 15-16):

“These plots are intended for within-sensor, qualitative comparison of spatial patterns and are not used for cross-sensor quantitative comparison.”

5. The parameters for the optical tweezer calibration are not sufficiently described. Specifically, the loading rate is a critical parameter that strongly influences protein unfolding forces. To ensure physiological relevance, the loading rate should mimic that experienced in the cellular environment, which recent work suggests is in the range of ~ 1 pN/s (see Hu et al., Cell, 187, 3445, (2024)). The authors should report their loading rate and discuss how it compares to the physiological regime, as this impacts the interpretation of the sensor's working force range.

We apologize for the omission and thank the reviewer for pointing out this critical parameter. In our optical tweezer experiments, we utilized a pulling speed of 100 nm/s, which corresponds to a loading rate of approximately 8-10 pN/s in the force range where the sensor responds. We acknowledge that this rate is higher than the physiological loading rate of ~ 1 pN/s recently reported by Hu et al. (Cell, 2024). Importantly, under our experimental conditions the TS module exhibited highly reversible mechanical behavior, as shown by nearly identical loading and unloading curves with no detectable hysteresis (Supplementary Fig. 1F). This reversibility indicates that the conformational changes underlying the fluorescence response occur on a timescale faster than our loading protocol, suggesting that the sensor is not limited by rate-dependent transitions and should respond similarly under slower, physiological loading rates.

Because our calibration aims to define the force range over which fluorescence changes occur, the higher loading rate does not alter the interpretation of the TS working range in cells. We have added the loading

rate to the Methods section (page 17, lines 2-3) and expanded the Discussion to address its physiological relevance as follows (page 12, lines 8-14):

“Although some GFP-based constructs have been reported to exhibit loading rate-dependent unfolding in single-molecule studies (Dietz and Rief, 2004; Mickler et al., 2007), our optical tweezer measurements revealed that the TS module undergoes rapid and reversible conformational changes under the loading rate used in our calibration (8-10 pN/s, corresponding to a pulling speed of 100 nm/s). The force-extension curves showed no detectable hysteresis over repeated pulling-relaxation cycles (Supplementary Fig. 1F), indicating that the fluorescence-relevant conformational changes are reversible and occur on a timescale faster than our loading protocol.”

Reviewer #3 (Remarks to the Author):

SUMMARY:

Fujiwara et al present a novel type of molecular tension sensor that does not rely on Forster Resonance Energy Transfer (FRET) as many tension sensors do, making it more accessible for certain applications. The authors take advantage of a circularly permuted version of EGFP (cpEGFP) that displays a range of intensities dependent on the force exerted on the molecule containing the cpEGFP. This allows for measurement of forces in the range of 0-6 pN using a single fluorescent protein. Fujiwara et al test this novel tension sensor design within the known force sensitive molecules alpha-actinin and alpha-catenin both in cells in culture and in mouse tissues. The authors also tag the molecules of interest with a red fluorescent mCherry protein at the C-terminus to track sensor localization and concentration. The sensors can apparently detect a range of forces in high spatiotemporal resolution, offering a promising approach for force sensing that requires less cumbersome analysis and could be more amenable to in vivo use.

Overall, this work could be a good proof-of-concept of an alternative force sensing module. I have some concerns as to how the sensors are portrayed relative to state-of-the-art FRET sensors, the experimental techniques proving functionality, and the unclear processing of image data. These concerns must be addressed to be confident in the functionality of these new sensors.

We appreciate the reviewer for the constructive evaluation of our work. In response to the concerns raised, we have performed additional experiments, revised and expanded the figures, and clarified the image-processing and methodological descriptions. These revisions have substantially improved the clarity and overall quality of the manuscript.

MAJOR COMMENTS:

- Some of the statements made about FRET-based tension sensors are factually incorrect. I agree that this proposed alternative has some advantages, but the state-of-the-art needs to be portrayed accurately. Here are the topics that should be addressed:

We sincerely thank the reviewer for carefully identifying these factual inaccuracies and for acknowledging the advantages of our sensors. We appreciate the opportunity to correct our description and have revised the Introduction accordingly, as addressed in detail for each point below.

o While the original vinculin tension sensor was generated with a spider silk (GPGGA)_n linker, there have been many other linkers used that result in different extension properties of the sensors, including increases in the dynamic range [<https://pmc.ncbi.nlm.nih.gov/articles/PMC6053308/>].

We thank the reviewer for pointing out this important clarification. In the revised Introduction, we have corrected our description of FRET-based tension sensors as follows (page 3, lines 9-11):
“The original vinculin TS used a spider-silk-derived (GPGGA)_n linker (Grashoff et al., 2010), and subsequent work has introduced additional extensible linkers with distinct mechanical properties and expanded dynamic ranges (LaCroix et al., 2018).”

o While intermolecular FRET can theoretically be an issue, it can and should be quantified by simultaneously expressing constructs with individual fluorophores so that the risk can be tested before drawing conclusions [<https://pubmed.ncbi.nlm.nih.gov/30523675/>]. In the absence of intermolecular FRET, differences in tension sensor expression should have no impact on FRET signal.

We thank the reviewer for highlighting this important point. In response, we performed additional experiments to quantify potential intermolecular FRET with our sensors. We measured FRET efficiency in (i) MDCKII cells transiently overexpressing the α Actinin tension indicator and (ii) cardiomyocytes

isolated from the α Actinin tension indicator mice. As shown in Supplementary Fig. 4E and F, a weak FRET signal was detectable only under conditions of artificially high overexpression in MDCKII cells, whereas no measurable FRET was observed at physiological expression levels in either MDCKII cells or cardiomyocytes. These results have been incorporated into the revised manuscript (page 9, lines 13-20) and demonstrate that intermolecular FRET is negligible under physiological and in vivo conditions, consistent with the reviewer's point that expression-level differences do not affect the readout in the absence of intermolecular FRET.

o The amount of FRET TS proteins is not uncertain. The concentration of the FRET sensor is measured by imaging the acceptor fluorophore alone with its excitation and emission wavelengths. Fluorescence of the acceptor is not dependent on energy transfer, and this approach has been used in most FRET TS papers to measure sensor concentration, including those cited by the authors.

We thank the reviewer for this important clarification. We fully agree that the abundance of FRET-based tension sensors can be quantified by direct excitation of the acceptor fluorophore. To reflect this point accurately, we have revised the Introduction. The relevant sentence now reads (page 3, lines 19-21):
“3) although direct acceptor excitation can quantify TS abundance, accurately comparing TS levels in heterogeneous in vivo tissues is challenging due to differences in expression levels and optical properties among cell types.”

This revision acknowledges that TS abundance can indeed be measured, while also clarifying that heterogeneity in in vivo tissues complicates quantitative comparisons of FRET signals across different cellular contexts.

o FRET is not “simply based on the distance between two independent fluorescent proteins”, it is also dependent on relative orientation of the fluorescent protein dipoles. This may actually help the argument for the use of cpEGFP if the authors are confident that changes in fluorescence of cpEGFP are only due to distance between the GFP domains.

We thank the reviewer for this important clarification. We fully agree that FRET efficiency is governed not only by donor–acceptor distance but also by the relative orientation of their transition dipoles, and we have revised the Introduction to reflect this point. The revised text now explicitly notes the orientation dependence of FRET (page 3, lines 17-18). We also appreciate the reviewer's suggestion that this complexity further underscores the conceptual appeal of our cpEGFP-based single-fluorophore approach.

o FRET-based tension sensors have been successfully used in vivo. I have listed here only a few examples. [<https://pmc.ncbi.nlm.nih.gov/articles/PMC11874908/>,
<https://www.nature.com/articles/s41598-023-50142-z>,
<https://journals.plos.org/plosbiology/article?id=10.1371/journal.pbio.3000057>].

We thank the reviewer for this helpful clarification and for providing additional examples of successful in vivo applications of FRET-based tension sensors. We fully agree with the reviewer's point. In the revised Introduction, we now explicitly acknowledge the successful in vivo use of FRET-based TSs and have incorporated all references suggested by the reviewer, in addition to representative prior studies (page 3, lines 13-16).

- The authors make heavy use of the myosin II inhibitor blebbistatin to demonstrate the functionality of their sensors. Blebbistatin is blue-light sensitive and becomes phototoxic upon stimulation with the wavelength of light required to excite EGFP. Additionally, blebbistatin is highly fluorescent in aqueous solutions greater than 10 μ M. It is not used in live cell imaging for these reasons. There are alternatives that have been recently developed, such as para-aminoblebbistatin [<https://pmc.ncbi.nlm.nih.gov/articles/PMC4886532/>]. If the reagent used in this work is a different form of blebbistatin, this should be clearly stated in the methods. If the authors used regular blebbistatin, I

hesitate to draw conclusions from any experiments involving the application of blebbistatin to live cells and imaged using blue light.

We thank the reviewer for raising this important concern regarding the blue-light sensitivity and phototoxicity of blebbistatin. We fully agree that conventional blebbistatin can cause artifacts under EGFP excitation.

To address this issue directly, we repeated the key assays using para-aminoblebbistatin (p-AmBleb), a photostable and non-fluorescent myosin II inhibitor (Várkuti et al., Sci. Rep., 2016). We performed parallel experiments in MDCKII cells expressing either the α Act TS or α Act-EGFP control (Supplementary Fig. 2C and D) and (ii) NIH3T3 cells expressing either the α Cat TS or α Cat-EGFP control (Supplementary Fig. 3B and C). As reported by Várkuti et al., p-AmBleb has reduced cell permeability and therefore requires higher concentrations and a slower onset. Consistent with this, blebbistatin (15 μ M) induced significant relaxation within <5 min, whereas p-AmBleb (30 μ M) required ~14 min to reach a comparable effect. Importantly, despite these kinetic differences, p-AmBleb produced the same qualitative outcomes: both α Act and α Cat tension indicators showed clear increases in TS fluorescence and relaxation ratio, whereas the corresponding EGFP controls showed no change. No photobleaching or toxicity was observed (Supplementary Fig. 2E). These results demonstrate that the tension-dependent fluorescence changes reported by our TS module are independent of blebbistatin's photolability. The new p-AmBleb data have been incorporated into the revised manuscript.

- Fluorescent proteins are easily photobleached under typical imaging conditions, especially when repeatedly imaged over time. Different fluorescent proteins bleach at different rates, which can make it difficult to measure FRET during a time series. It is imperative to show that photobleaching is not affecting the timelapse measurements. While it is promising that the untreated alpha-actinin TS and mCherry fluorescence stays stable in Figure 1f, there seems to be a large drop in fluorescence in the first time points in Figure 4f. One way to accomplish this would be to perform an experiment on fixed samples expressing the tension sensors with the same imaging parameters used in the various timelapse experiments and show that relaxation ratio is unaffected by repeated imaging.

We thank the reviewer for raising this important point. To determine whether photobleaching could influence our time-lapse measurements, we performed bleaching-control experiments using fixed MDCKII cells overexpressing the α Act tension indicator. When imaged under the identical laser power, exposure, and frame rate used in our blebbistatin and p-AmBleb assays—and for more than twice the duration—we observed no detectable decrease in either the TS (green) or mCherry (red) fluorescence (Supplementary Fig. 1D and 2E). The relaxation ratio likewise remained stable, demonstrating that our imaging conditions do not induce measurable bleaching.

Regarding the fluorescence drop at the beginning of Fig. 4F, this effect does not arise from photobleaching. These particular experiments were performed using the Dragonfly confocal system mounted on an Olympus microscope, whose Z-drift stabilization is less precise than the Nikon Perfect Focus System used in other assays. Upon drug addition, a transient axial (Z) drift occurred, momentarily reducing both green and red fluorescence intensities. Importantly, this effect was absent in all Nikon-based imaging where axial drift is automatically corrected by the Perfect Focus System.

Together with the fixed-sample controls, these results demonstrate that the early intensity decrease in Fig. 4F reflects a transient focal shift rather than photobleaching, and that bleaching does not affect the interpretation of our live-cell time-lapse imaging.

- Is the data in Figure 1f, S1d, 2a, etc being normalized in some way to start at 100 a.u.? Is this obscuring variability in the starting fluorescence levels? This is especially relevant for comparing the different controls in 2a – I would assume Control-3 would start out at a higher EGFP intensity than the original sensor. It is important to be able to use a tension sensor to read out differences in tension levels across groups, not just in a single group in response to a treatment.

We fully agree with the reviewer that normalization can, in principle, obscure variability in starting fluorescence levels, and such baseline differences would indeed be relevant if absolute tension were inferred directly from baseline intensity. However, in our experimental system, such baseline comparisons are inherently unreliable. Because transient transfection produces substantial variability in expression levels, baseline fluorescence primarily reflects expression level rather than tension, and identical exposure settings would either saturate bright cells or underexpose dim cells. For this reason, exposure time was adjusted per visual field, and fluorescence values were normalized to the intensity at drug addition ($F_0 = 100$) solely to visualize temporal dynamics, as detailed in the Discussion (page 21, line 26 – page 22, line 8)

For transparency and to illustrate the actual processing, we have included both normalized and raw, non-normalized fluorescence traces for all cells expressing the α Act TS and the EGFP control in Supplementary Fig. 2B. As expected, the EGFP control (Control-2) is brighter than the α Act TS, whereas Δ ABD α Act TS control (Control-3) exhibit comparable baseline intensity. These baseline differences arise from expression variability and construct differences, rather than from differences in molecular tension. Therefore, baseline intensity cannot be used to compare absolute tension across constructs in this experimental context.

Importantly, our tension readouts do not rely on absolute fluorescence values but on relative changes from each cell's own baseline, such as drug-induced relaxation or time-dependent changes. Thus, although basal fluorescence cannot report basal tension in this system, the interpretation of relative tension changes is unaffected by the normalization, and comparisons of these relative responses—i.e., how strongly each construct responds to a perturbation— across groups remain valid.

MINOR COMMENTS:

- FRET can occur between EGFP and mCherry [<https://www.nature.com/articles/s41598-023-50142-z>, <https://journals.plos.org/plosone/article?id=10.1371/journal.pone.0001916>]. As alpha-actinin functions as an antiparallel dimer, it is possible that intermolecular FRET occurs between the EGFP of one molecule and the mCherry of another. It would be best practice to perform an intermolecular FRET control by co-expressing alpha-actinin-cpEGFP and alpha-actinin-mCherry.

Thank you for this insightful comment. We agree that potential intermolecular FRET must be evaluated. As described in our response to major comment #1-2 (and now included in Supplementary Fig. 4E and F), we directly quantified intermolecular FRET by measuring FRET efficiency in both transiently overexpressing MDCKII cells and cardiomyocytes expressing the α Actinin TS indicator at physiological levels. A weak FRET signal was detectable only under artificial overexpression conditions in MDCKII cells, whereas no measurable intermolecular FRET was observed at physiological or in vivo expression levels. These results confirm that intermolecular FRET is negligible in the conditions used in this study.

- Alpha-actinin more specifically localizes to actin filaments, not just “the cytosol” as stated in line 20.

Thank you for pointing this out. We agree with the reviewer and have corrected the wording to indicate that α -actinin localizes to actin filaments rather than “the cytosol.” The revised sentence now reads (page 5, line 21):

“These molecules localize to distinct subcellular regions, such as actin filaments and adherens junctions (Supplementary Fig. 1A).”

- The talin cpEGFP tension sensor was disregarded due to the weak fluorescence, but is that not indicative of high levels of tension? Does signal get restored with blebbistatin (or other cytoskeletal inhibitor) treatment? I would be interested in following up on this finding. However, if this sensor is really not going to be explored further in this manuscript, I would leave it out entirely.

Thank you for the comment. In the talin construct, both the cpEGFP-based TS signal and the C-terminal mCherry fluorescence were too weak for reliable detection. As this prevented meaningful analysis, and in line with the reviewer's suggestion, we have removed the talin tension sensor from the manuscript.

- Please explain what you mean by “the fluorescence of the alpha-actinin tension indicator changed from orange to yellow-green”. It needs to be clear that you are talking about an overlay/merge (I assume) of the mCherry and EGFP signals. Please also explain the “relaxation ratio” in bit more detail and how it can be interpreted.

Thank you for pointing this out. We now clarify in the revised manuscript that the “orange” to “yellow-green” change refers to the merged images of the green (cpEGFP-based TS module) and red (mCherry) channels. The shift in merged color reflects the relative green/red fluorescence, with more yellow-green indicating lower tension and more orange-red indicating higher tension.

We also expanded the description of the “relaxation ratio.” The relaxation-ratio images represent the pixel-wise green/red (G/R) fluorescence ratio visualized using a cool-warm colormap. Higher ratios (warm colors) indicate reduced tension, whereas lower ratios (cool colors) indicate higher tension. These ratio images provide a quantitative and more visually apparent representation of tension changes than the merged images. The revised text now explains both points in figure legends and Results section (page 7, lines 2-3 and 5-7).

- I don't understand in Figure 3d how both the relaxation ratio and the tension ratio range from 0-3 if they are just inverse of each other. Is there something else going on in this calculation? Since this is the only place that tension ratio is used, I would suggest leaving it out and just inverting the color map for the relaxation ratio. Unless something more complicated is going on that is not clearly explained.

Thank you for pointing this out. The relaxation ratio (green/red) and tension ratio (red/green) should indeed be mathematical inverses. In our original figure, both appeared in the 0–3 range due to subsequent normalization steps applied for visualization, which was not clearly explained and caused unnecessary confusion. Following the reviewer's suggestion, we have removed the tension ratio and now present only the relaxation ratio (Fig. 3E and F).

- Is the data in Figure 3f from a single fibroblast not in contact with other cells? In this scenario, how is alpha-catenin under any tension without the existence of adherens junctions?

Thank you for this comment. The cell shown in Fig. 3F is indeed not in contact with neighboring cells. Although α Catenin is best known for its role at adherens junctions, several studies have shown that α Catenin can also associate with the actin cytoskeleton independently of cadherin-based adhesion (Drees et al., Cell 2005, PMID: 16325583; Benjamin et al., J Cell Biol 2010, PMID: 20404114). NIH3T3 fibroblasts generate substantial actomyosin contractility, and we consider that this intracellular actin tension is sufficient to load α Catenin in the absence of cell–cell contacts.